# Crumbs2 mediates ventricular layer remodelling to form the spinal cord central canal

Christine M. Tait[1,�‡a], Kavitha Chinnaiya[1,�], Elizabeth Manning[1], Mariyam Murtaza[1,‡b], John-Paul Ashton[1], Nicholas Furley[1], Chris J. Hill[1], C. Henrique Alves[2,‡c], Jan Wijnholds[2], Kai S. Erdmann[1], Andrew Furley[1], Penny Rashbass[1], Raman M. Das[4,‡d], Kate G. Storey[4], Marysia Placzek[1]*

**1** Department of Biomedical Science and Bateson Centre, University of Sheffield, Sheffield, United Kingdom, **2** Department of Ophthalmology, Leiden University Medical Centre, Leiden, the Netherlands, **3** Netherlands Institute for Neuroscience, KNAW Amsterdam, Amsterdam, the Netherlands, **4** Division of Cell and Developmental Biology, School of Life Sciences, University of Dundee, Dundee, United Kingdom

These authors contributed equally to this work.
‡a Current address: Berkshire Royal Hospital, Berkshire, United Kingdom
‡b Current address: Menzies Health Institute Queensland, Griffith University, Queensland, Australia
‡c Current address: Coimbra Institute for Clinical and Biomedical Research, Faculty of Medicine and Center for Innovative Biomedicine and Biotechnology, University of Coimbra, Coimbra, Portugal
‡d Current address: Division of Molecular and Cellular Function, University of Manchester, Manchester, United Kingdom
* m.placzek@sheffield.ac.uk

**Data Availability Statement:** All relevant data are within the paper and its Supporting Information files.

## Abstract

In the spinal cord, the central canal forms through a poorly understood process termed dorsal collapse that involves attrition and remodelling of pseudostratified ventricular layer (VL) cells. Here, we use mouse and chick models to show that dorsal ventricular layer (dVL) cells adjacent to dorsal midline Nestin[(+)] radial glia (dmNes[+]RG) down-regulate apical polarity proteins, including Crumbs2 (CRB2) and delaminate in a stepwise manner; live imaging shows that as one cell delaminates, the next cell ratchets up, the dmNes[+]RG endfoot ratchets down, and the process repeats. We show that dmNes[+]RG secrete a factor that promotes loss of cell polarity and delamination. This activity is mimicked by a secreted variant of Crumbs2 (CRB2S) which is specifically expressed by dmNes[+]RG. In cultured MDCK cells, CRB2S associates with apical membranes and decreases cell cohesion. Analysis of $Crb2^{F/F}/Nestin\text{-}Cre^{+/-}$ mice, and targeted reduction of $Crb2$/CRB2S in slice cultures reveal essential roles for transmembrane CRB2 (CRB2TM) and CRB2S on VL cells and dmNes[+]RG, respectively. We propose a model in which a CRB2S–CRB2TM interaction promotes the progressive attrition of the dVL without loss of overall VL integrity. This novel mechanism may operate more widely to promote orderly progenitor delamination.

## Introduction

The ventricular layer (VL) of the embryonic spinal cord is composed of pseudostratified radial glial stem/progenitor cells that line the central lumen. VL cells express the SRY-related HMG-

**Funding:** This research has been supported by grants fromthe UK Medical Research Council (https://mrc.ukri.org/; G0401310 to MP, the Wellcome Trust (https://wellcome.ac.uk/; 212247Z18Z to MP; 102817A1A to KGS; 208401 to KGS) and The European Union (https://ec. europa.eu/research/fp7/index_en.cfm; HEALTH F2-2008-200234 to PR and JW and The Netherlands Organisation for Health Research and Development (https://www.zonmw.nl/en/; ZonMw 43200004 to JW). The funders had no role in study design, data collection and analysis, decision to publish, or preparation of the manuscript.

**Competing interests:** The authors have declared that no competing interests exist.

**Abbreviations:** α-M2, anti-mouse astrocyte-surface antigen; aPKC, atypical protein kinase C; BMP, bone morphogenetic protein; Cdc42, cell division control protein 42; CNS, central nervous system; CRB, Crumbs; CRB2/Crb2, Crumbs2; CRB2S, secreted CRB2; CRB2TM, transmembrane CRB2; DF, dorsal funiculus; dmNes+RG, dorsal midline Nestin$^{(+)}$ radial glia; dVL, dorsal ventricular layer; EL, ependymal layer; EM, electron microscopy; F, filamentous; GADPH, glyceraldehyde 3-phosphate dehydrogenase; GFP, green fluorescent protein; HH, Hamburger-Hamilton; MDCK, Madin-Darby kidney cells; NKX6.1/Nkx6.1, NK6 homeobox 1; PAR3/Par3, polarity protein PAR3; PAX6/Pax6, paired-box protein 6; PS980-Baz, Bazooka; RFP, red fluorescent protein; Rok, Rho-associated protein kinase; Shh, Sonic hedgehog; SOXB1, SRY-related HMG-box B1 transcription factors; SOX2/Sox2, SRY-related HMG-box 2; SVL, subventricular layer; VF, ventral funiculus; VL, ventricular layer; vVL, ventral ventricular layer; ZO-1/Zo-1, Zona occludens 1.

box transcription factors, SOXB1 [1], a feature of their neuroepithelial origin [2,3], and differentially express homeodomain transcription factors, a feature of dorsoventral patterning [4–6]. In early embryogenesis, VL cells undergo neurogenesis in a process that involves apical constriction, adherens-junction disassembly, acto-myosin-mediated abscission and mediolateral migration [7,8]. Following the major period of neurogenesis, VL cells switch to gliogenesis, and glial cells migrate out of this layer [9–12].

Concomitant with the transition to gliogenesis (around E12 in the mouse) the VL begins to remodel, ultimately giving rise to the ependymal layer (EL) surrounding the adult central canal. Ependymal cells constitute key components of a quiescent stem cell niche [13–15], are implicated in glial scar formation after spinal injury [15–17], and serve important mechanical and sensory functions [18,19]. Multiple steps contribute to the remodelling of the VL into the EL [1,20], including dorsal collapse, a delamination of dorsal ventricular layer cells (hereafter termed dVL cells) that results in a pronounced dorsoventral reduction in the length of the lumen [20–28]. Little is understood, however, of the mechanisms that mediate dorsal collapse.

Proteins of the Crumbs (CRB) and PAR complexes (the latter composed of PAR3/PAR6/ aPKC) are present on the apical side of epithelial cells. In invertebrates, PAR- and CRB-complex proteins directly interact to determine the apicobasal axis and the position and stability of cell–cell adherens junctions [29–36]. PAR and CRB complex components are evolutionarily conserved and similarly regulate polarity, integrity, and morphogenesis of vertebrate epithelia, including the neural tube neuroepithelium [37–39]. In the mouse, *Crb2* (one of the three vertebrate *Crb* genes) is required for maintenance of the apical polarity complex [40–42], and in zebrafish, the two *crb2* genes have been implicated in retinal organisation. *Crb2a* (*oko meduzy*, *Ome*), in particular, was described as a determinant of apicobasal polarity in the retina, and its loss caused severe basal displacement of cell junctions in neuroepithelial cells [43,44]. Intriguingly, the finding in zebrafish that mutation in *pard6yb* results in the failure of dorsal collapse [45,46] suggests that apical polarity complex regulation plays a critical role in VL remodelling. However, it remains unclear whether and how components of the apical polarity complex change during dorsal collapse, nor in which cell populations these proteins are regulated and required, nor how they regulate downstream effectors of epithelial integrity.

The roof plate is a specialised glial cell population that defines the dorsal neural tube/spinal cord midline and patterns dorsal neuroepithelial cells [47]. In mid-embryogenesis, roof plate cells are transformed from wedge-shaped cells into a thin, dense septum of elongated dorsal midline Nestin$^{(+)}$ radial glia (hereafter termed dmNes+RG) that extend from the ventricle to the pia [20–23,26,45,48] and eventually contribute to the EL [20,28,49]. Elongation of the roof plate/dmNes+RG occurs reciprocally with reduction of the lumen [20,23,26,28,45,50], and dmNes+RG are known to play a critical role in dorsal collapse. Studies in zebrafish suggest that the filamentous (F)-actin cytoskeleton belt that defines the apical side of VL cells, and whose constriction drives early neurulation [51–54] and neuronal delamination [7,8], is also required for dorsal collapse: this involves a process that depends on its appropriate tethering by elongating roof plate/dmNes+RG cells [45,50]. However, in addition to their tethering function, dmNes+RG cells may play an active role in promoting delamination by locally regulating VL cell polarity.

Here, we provide evidence that in mouse and chick, Crumbs2 (CRB2) proteins are required for dorsal collapse. In particular, we show that a CRB2-mediated interaction of dmNes+RG cells and adjacent dVL cells drives the progressive delamination of dVL cells and the transformation of the VL to the EL. Expression of apical/tight junction components and adhesion complexes are maintained at high levels on the apical end-feet of dmNes+RG throughout lumen diminution but are dramatically reduced on dVL cells as they delaminate. dmNes+RG are rich in secretory vesicles, and gain-of-function in vivo studies show that they secrete a

factor that promotes progenitor cell delamination. Furthermore, they express a variant of CRB2 that can be secreted (CRB2S) and appears to mediate this activity. Ex vivo cell culture studies show that CRB2S binds to the apical surfaces of CRB2-expressing epithelial cells and reduces their polarity and cohesion. In vivo analysis of *Crb2*$^{F/F}$/*Nestin-Cre*$^{+/-}$ mice and targeted reduction of *Crb2*/CRB2S in dmNes$^+$RG in slice cultures reveal essential roles for transmembrane CRB2 (CRB2TM) and CRB2S on VL cells and dmNes$^+$RG, respectively. In this interaction, CRB2 proteins are required to remove rather than maintain cells within an epithelium. We propose that collapse is initiated by the release of CRB2S from dmNes$^+$RG that acts on CRB2TM-expressing VL cells, causing down-regulation of polarity and junctional proteins and their decreased cohesion. We suggest that these are a critical step in the exclusion of these cells from the VL and transformation of the VL to the EL. Our findings suggest a model in which CRB2S acts cell non-autonomously to orchestrate progenitor cell delamination from an epithelium through a mechanism that retains epithelial integrity.

## Results

### Collapse occurs through attrition of dVL cells

Dorsal collapse of the mouse spinal cord occurs over the period E14–E17. At thoracic levels, the lumen spans almost the entire dorsoventral length of the spinal cord at E14 and reduces to a two-fifth span at E15 and a one-fifth span by E17 (Fig 1A–1E and 1P). As described recently [1], dorsal collapse is predicted by differences in VL morphology. In the ventral ventricular layer (vVL), the lumen is narrow and nuclei are tightly packed and mediolaterally oriented (Fig 1A–1D, 1P, S1A and S1A′ Fig), whereas in the dVL, the lumen is wide and nuclei are more loosely arranged (Fig 1A–1C and 1P, S1A and S1A′ Fig). The dVL reduces significantly in length on each consecutive day during collapse (Fig 1A–1D and 1P), whereas the vVL does not (Fig 1A–1D and 1P). Dorsal collapse is the most obvious remodelling event but is mirrored ventrally by a rearrangement of floor plate cells, only a subset of which remain within the central canal (Fig 1B arrowhead and see [1]).

SOXB1 proteins, namely SOX1, SOX2 and SOX3, are expressed on all VL cells during remodelling (Fig 1F–1I and S1C–S1F Fig). Quantitative analyses show a reduction in the number of SOX2$^{(+)}$ cells in the VL during collapse (Fig 1J) and see [1]; excluded cells continue to express SOX1-3 (Fig 1G and 1H and S1C–S1F Fig). Likewise, the paired-box protein (PAX6), which marks cells in all but the ventral-most VL region (consistent with its earlier expression in progenitor subsets [55]) reveals a similar proportional reduction (S1G–S1J Fig). By contrast, the homeobox protein NKX6.1, which marks ventral progenitor subsets [55], is restricted to vVL cells at E14 (Fig 1K) and is then detected at the ventral midline as the floor plate pinches off (Fig 1L–1N). There is less proportional reduction in the number of Nkx6.1$^{(+)}$ VL cells (Fig 1O). Together, these observations are consistent with the idea that dorsal collapse is largely driven through the attrition of the dVL [1,24].

### dVL cells reorientate during attrition

Consistently, during dorsal collapse, we detect a nuclear bridge spanning the two sides of the dVL. Regardless of the stage, the bridge is detected 2–8 cell diameters below the dorsal-most lumen (Fig 2A–2D arrowheads). As VL cells dorsal to the bridge are excluded, their nuclei appear to reorient from mediolateral to dorsoventral (S1A and S1A′ Fig). By the end of dorsal collapse, all nuclei dorsal to the vVL are dorsoventrally elongated (S1B and S1B′ Fig). Together, these results begin to suggest a remodelling of dVL cells during collapse.

To better characterise the position of reorientating nuclei, we compared expression of SOX2 to that of Nestin. From E14.5, the dorsal and ventral midline are characterised by

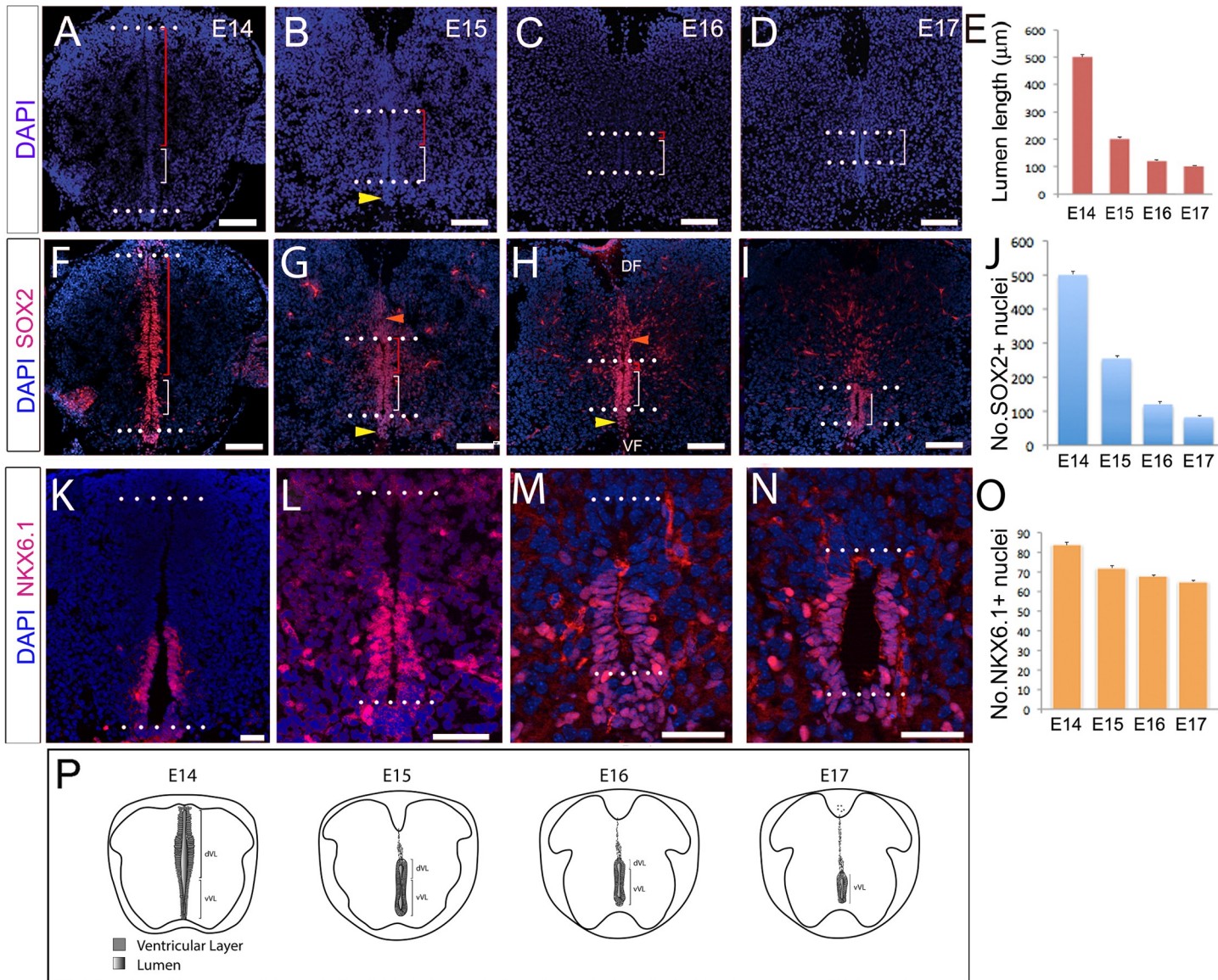

**Fig 1. Collapse occurs through attrition of dVL cells.** Transverse sections through the spinal cord of E14 (A,F,K), E15 (B,G,L), E16 (C,H,M), and E17 (D,I,N) mouse embryos. (A-D) DAPI labelling shows diminution of dVL: dotted lines show upper and lower limits of lumen; red bracket indicates dVL; white bracket indicates vVL. (E) Quantitative analysis of lumen length. (F-I) Immunolabelling shows SOX2(+) cells throughout the VL at E14 (F), then (G-I) in the VL (between dotted lines) and excluded dorsal to the VL (orange arrowheads) or dissociated ventral to the VL (yellow arrowheads), and many in the midline near the ventral funiculus (VF) and dorsal funiculus (DF). (J) Quantitative analysis shows reduction of SOX2(+) VL cells around the lumen. (K-N) NKX6.1 marks the vVL, and many NKX6.1(+) progenitors are retained in collapse (shown quantitatively in (O)). Underlying data can be found in S1–S3 Tables. (P) Schematic showing diminution of the lumen over E14–E17, and position of dVL and vVL cells. Scale bars: A-D; F-I: 100 μm; K-N: 50 μm. DF, dorsal funiculus; dVL, dorsal ventricular layer; NKX6.1, NK6 homeobox 1; SOX2, SRY-related HMG-box 2; VF, ventral funiculus; VL, ventricular layer; vVL, ventral ventricular layer; (+), expressing.

Nestin(+) radial glial cells (Fig 2E–2I, S2 Fig, and see [1,20–23,26,28,45,48]). dmNes+RG, which elongate as collapse proceeds, are thought to tether the diminishing VL to the pial surface [45]. Double-labelling of Nestin and SOX2 shows that the dorsal-most pole is a nuclei-free area occupied by dmNes+RG end-feet (Fig 2J–2L″) and confirms that the nuclear bridges below the dmNes+RG end-feet/dorsal lumen are SOX2(+) (Fig 2J–2L arrowheads). Towards the end of dorsal collapse (E17.5), a bright spot of Nestin immunoreactivity is detected in the SOX2(+) bridge (Fig 2M and 2M′) suggesting a physical interaction of dmNes+RG and dVL cells.

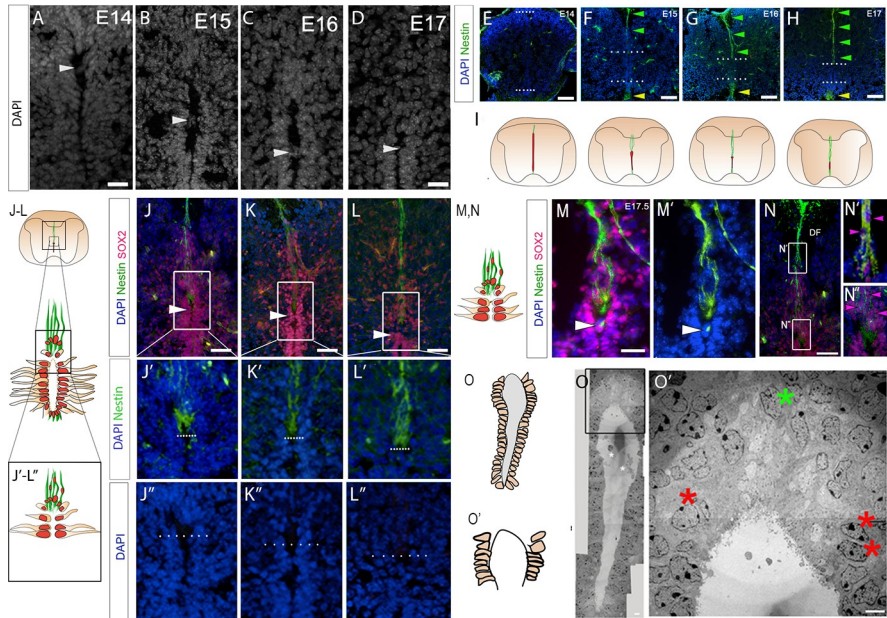

**Fig 2. Cell remodelling during dorsal collapse.** Transverse sections through the spinal cord of E14–E17 mouse embryos. (A-D) High-magnification views of dVL/lumen; arrowheads point to nuclear bridges. (E-H) Immunolabelling reveals dmNes+RG that elongate over E14–E17 (green arrowheads). Shorter Nestin(+) radial glial cells are also detected along the ventral midline (yellow arrowheads). Dotted lines indicate ends of lumen. (I) Schematic showing midline radial glia (green) and lumen (red). As dmNes+RG lengthen over E14–E17, the VL shortens. (J-L) Co-labelling of Nestin and SOX2 at E15 (J), E16 (K), and E17 (L). Regions analysed are indicated by large box in schematic (J-L). Small box in schematic (J-L) and white box in sections (J-L) show regions analysed in (J′-L″). (J′-L″) High-magnification views of Nestin and DAPI (J′-L′) or DAPI alone (J″-L″) reveal that the dorsal-most lumen is occupied by the end-feet of dmNes+RG. Dotted lines in J-L″ indicate dorsal lumen; arrowheads in J-L indicate SOX2(+) nuclear bridge. (M-N″) Double labelling of Nestin and SOX2 at E17.5. In addition to expression on dmNes+RG, Nestin can be detected in the SOX2(+) nuclear bridge below the dorsal lumen (M, M′ arrowhead). Dorsoventrally oriented SOX2(+) nuclei are closely apposed to dmNes+RG, both in the region of the dorsal funiculus (N, N′) and around the dmNes+RG endfeet (N, N″). Arrowheads point to representative SOX2(+) nuclei. (O) EM image at E15.5: boxed region shown in O′. Red asterisks show VL nuclei that are close to but do not abut the lumen; green asterisk shows nuclei that are dorsoventrally oriented in the dorsal midline some distance above the lumen. Schematics show regions shown in (O) and (O′). Scale bars: A–D, J′–L″, M, M′, 20 μm; E–H, 100 μm; J–L, N, 50 μm; O, 10 μm; O′, 1 μm. DF, dorsal funiculus; dmNes+RG, dorsal midline Nestin(+) radial glia; dVL, dorsal ventricular layer; EM, electron microscopy; SOX2, SRY-related HMG-box 2; VL, ventricular layer.

Double-labelling of Nestin and SOX2 confirms, additionally, that many SOX2(+) cells are closely associated with dmNes+RG and can be detected as far away as the dorsal funiculus (Fig 2N–2N″).

To validate the arrangement/reorganisation of dVL cells, we performed transmission electron microscopy (EM) imaging at E15.5–E16 ($n$ = 3 embryos; 6 sections), a time when dorsal collapse is underway. The narrower vVL and wider dVL zones can be distinguished in EM images (Fig 2O). Quantitative analyses confirmed that the looser arrangement of dVL cells to vVL cells (21.5 ± 1.04 nuclei/100μm$^2$ and 54.0 ± 1.83 nuclei/100 μm$^2$ respectively) is significantly different ($p < 0.0001$; S4 Table), and high magnification views confirm that the dorsal-most pole lacks nuclei (Fig 2O′). Notably, however, although nuclei in the dVL are loosely arranged, they remain abutted to the lumen except for the 2–3 nuclei directly adjacent to the nuclei-free dorsal pole. These appear to have moved away (Fig 2O′ red asterisks), suggesting an organised delamination of dorsal-most dVL cells.

In summary, a first indication of dorsal collapse is a local cell reorganisation and remodelling of the dVL. Key changes include formation of a bridge just below the dorsal-most pole,

the re-orientation of nuclei dorsal to the bridge, and distancing of nuclei from the lumen in cells that are immediately adjacent to the endfeet of dmNes$^+$RG.

## Dorsal collapse involves cell delamination and ratcheting

To better examine the rearrangement of cells during dorsal collapse, we performed time-lapse imaging of spinal cord slice cultures, focusing on the dorsal region of the collapsing spinal cord. After transfecting sparse numbers of cells with membrane-GFP (green fluorescent protein) and histone-RFP (red fluorescent protein) ($n$ = 3 slices from 3 embryos; 6–20 dorsal cells electroporated in each), the dmNes$^+$RG cell can be detected because of its typical morphology and midline position (Fig 3A and 3A′, and S1 Movie). As predicted from our static studies (Fig 2), dmNes$^+$RG cells remain in a dorsal midline position throughout imaging (Fig 3A′ red arrows and red cells). At their apical ends, dmNes$^+$RG cells are closely apposed, on each side to a dVL cell (Fig 3A and 3A′ pink arrows and pink cells). These cells delaminate and, on each side, a second dVL cell becomes closely apposed to the dmNes$^+$RG cell (Fig 3A′ blue cells). These second dVL cells in turn delaminate (Fig 3A and 3A′ blue arrows and blue cells). Together this suggests that dorsal collapse proceeds through the progressive delamination of dVL progenitors that are immediately adjacent to the dmNes$^+$RG cell. Analysis of slices where higher numbers of cells were electroporated ($n$ = 2 slices; 20–30 dorsal cells electroporated) further suggests that post-delamination, dVL cells may actively migrate dorsally along the dmNes$^+$RG scaffold (S2 Movie and S3A Fig).

The proximity of electroporated cells, the fact that delaminating dVL cells can appear to cross one another (Fig 3A′ pink and blue cell on right-hand side), and the movement of cells in and out of focus, however, made it difficult to resolve the precise behaviour of cells, or the interaction of dmNes$^+$RG endfeet and apical-most parts of dVL cells in mouse slice cultures. We therefore performed similar studies in chick spinal cord slice cultures, in which it is easier to electroporate smaller numbers of cells, having first established that chicken embryos undergo collapse in vivo between E7 and E11 [56,57] (S4 Fig). Dorsal cells were targeted with membrane-GFP and time-lapse analyses performed ($n$ = 6 slices from 3 embryos; 6–12 dorsal cells electroporated in each). Cells behaved in a similar manner to that observed in mouse (Fig 3B and 3B′ and S3 Movie), but individual cell behaviours could be discerned. The dmNes$^+$RG cell elongates and thins (Fig 3B red arrow and Fig 3B′ red cell). A first and then a second dVL cell delaminate (Fig 3B pink and blue arrows and Fig 3B′ pink and blue cells). Intriguingly, we noted that in cases where it was possible to follow a dVL that stayed in focus throughout imaging, delamination of more distal dVL cells is preceded by a cell remodelling event: a protrusion 'reaches' towards the previous (delaminating) dVL cell or dmNes$^+$RG (Fig 3B and 3B′; blue arrowhead in Fig 3B; $n$ = 3 cells). Analysis of a small number of mouse cultures ($n$ = 2) in which only 1–2 dVL cells were electroporated revealed a similar reorientation prior to delamination ($n$ = 3 cells; S4 Movie). This remodelling may enable delamination and the dorsal migration of delaminating dVL cells to the top of the dmNes$^+$RG (Fig 3B and 3B′).

A single Z-plane view reveals that delamination is preceded by contact/close proximity between the dmNes$^+$RG cell and the dVL cell; thus, as dmNes$^+$RG endfeet extend down to them, dVL cells sequentially delaminate (Fig 3C and 3C′ and S5 Movie; additional evidence shown in S3B Fig and S6 Movie). Together, this suggests that dorsal collapse proceeds through the progressive delamination of dVL progenitors that are immediately adjacent to dmNes$^+$RG cell endfeet; as each dVL cell delaminates, the next dVL cell ratchets up and the dmNes$^+$RG endfoot ratchets down. This progressive series of interactions between two cell populations suggests a novel and specific delamination mechanism.

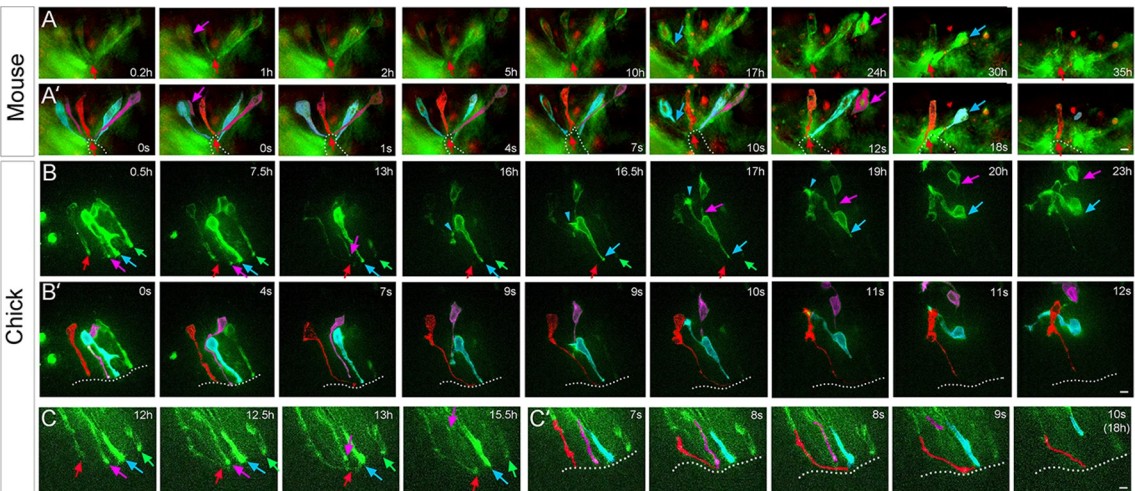

**Fig 3. Dorsal collapse involves cell delamination and ratcheting.** (A) Sequential stills from time-lapse imaging after electroporation of membrane-GFP histone-RFP into mouse. A midline cell whose morphology indicates it to be a dmNes⁺RG remains throughout the culture (red arrowhead; in total, 6 such cells observed from 3 slices). dVL cells adjacent to the dmNes⁺RG delaminate sequentially; on left-hand side, cells delaminate at 1 hour (pink arrow) and 17 hours (blue arrow); on right-hand side, cells delaminate at 24 hours (pink arrow) and 30 hours (blue arrow). (A′) Same images; cells colour-coded. In (A), timeframes refer to real time; in (A′), timeframes refer to S1 Movie. In total, *n* = 9 stepwise delaminating cells observed from 3 slices. (B) Sequential stills from time-lapse imaging after electroporation of membrane-GFP into chick. Arrows point to endfeet/apical parts of cells. dmNes⁺RG (red arrow) thins and elongates (0.5–13 hours; elongation observed in 5/6 slices). dVL cells adjacent to dmNes⁺RG sequentially delaminate (pink and blue arrows 13–16.5 hours; *n* = 10 cells observed in 5/6 slices). A distant dVL cell remains in situ (green arrow). The middle dVL cell (blue) reaches onto the dmNes⁺RG/previous dVL cell prior to delamination and dorsal migration (16.5–23 hours; blue arrowheads). (B′) Same images; cells colour-coded. Time frames in (B) refer to real time and in (B′) refer to S3 Movie. (C) A single Z-plane view showing end-feet/apical part of cells shown in (B). dmNes⁺RG endfoot (red arrow) contacts first dVL cell (pink arrow) at 12.5 hours, which delaminates at 13 hours. dmNes⁺RG endfoot contacts the second dVL cell (blue arrow) at 13 hours, which delaminates around 15.5 hours. (C′) Same images; cells colour-coded. Timeframes in (C) refer to real time and in (C′) refer to S4 Movie. dmNes⁺RG, dorsal midline Nestin⁽⁺⁾ radial glia; dVL, dorsal ventricular layer; GFP, green fluorescent protein; RFP, red fluorescent protein.

## Diminished adhesion junctions on VL progenitors adjacent to dmNes⁺RG

The stable retention of the dmNes⁺RG at the dorsal pole and delamination of adjacent dVL progenitors led us to predict that these two cell types will show distinctive profiles of apical adhesion and junction proteins that, in other systems, govern epithelial integrity and cell delamination. We therefore examined expression of Zona occludens 1 (ZO-1), a tight junction–associated protein, and the apical polarity proteins aPKC, CRB2, and PAR3 at E13.5 (prior to collapse), E15–E15.5 (maximal dorsal collapse), and E17 (termination of collapse). At the same time, we assayed expression of phalloidin, a marker of F-actin, previously suggested to anchor dmNes⁺RG and VL progenitor cells [45]. At E13.5, junction/adhesion proteins are detected in a continuum on the apical side of all VL progenitor cells (S5A–S5C Fig). By contrast, at E15, junction/adhesion proteins are maintained on vVL cells and on the end-feet of dmNes⁺RG, but ZO-1, CRB2, and PAR3 are barely detected on dVL cells (Fig 4A–4D, 4A′–4D′ and 4A″–4D″) and on dVL cells that lie immediately adjacent to the dmNes⁺RG, Zo-1, aPKC, and PAR3 appear completely absent (Fig 4A–4A″, 4B–4B″ and 4D–4D″ green arrowheads; *n* = 24–30 cells each, imaged from a minimum of 3 embryos; quantitative measurements shown in S5D Fig), and CRB2 is absent or reduced on these cells (Fig 4C–4C″; *n* = 24 cells from 8 embryos; S5 Fig). By E17, some dorsal discontinuity is still apparent, albeit less obvious (Fig 4F–4I and S5E Fig; *n* = 8 slices from 4 embryos), and by E18, apical/junctional proteins are again expressed as a continuum (S5F Fig). Phalloidin does not show the same marked absence but shows punctate labelling on the apical side of dVL cells at E15 (Fig 4E–

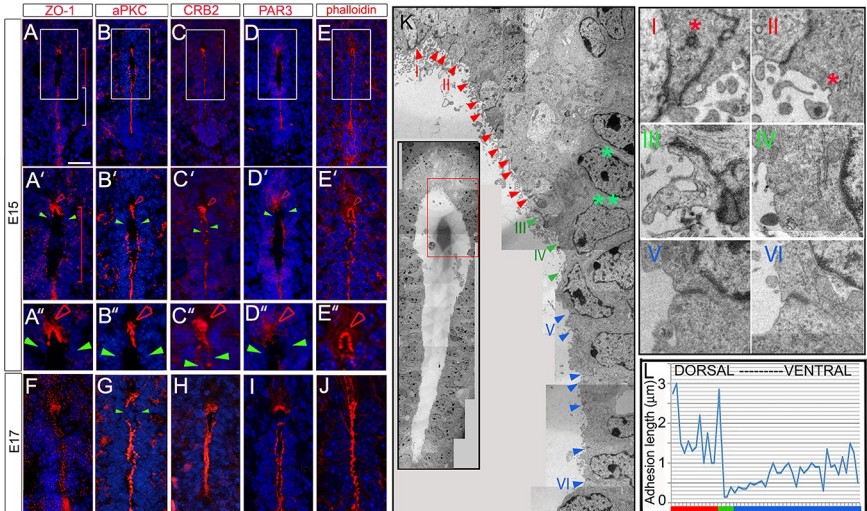

**Fig 4. Apical polarity proteins and tight junctions are reduced on delaminating VL cells.** (A-E″) Transverse sections through spinal cord of E15 mouse embryos immunolabelled as indicated. (A′-E′) show high-magnification views of boxed regions shown in (A-E); (A″-E″) show high-magnification views of regions indicated by arrowheads in (A′-E′). High expression of apical polarity proteins and the tight junction protein, ZO-1, are detected on the endfeet of dmNes+RG (open red arrowheads) but not detected (ZO-1, aPKC, PAR3) or barely detected (CRB2) on immediately adjacent dVL cells (green arrowheads). Phalloidin is expressed strongly on dmNes+RG and in a punctate manner throughout dVL cells. (F-J) At E17, junctional and polarity proteins likewise show reduced/no expression on VL cells that abut dmNes+RG, but phalloidin is detected in a continuum. White and red brackets show vVL and dVL regions, respectively. (K) EM images of the E15.5 VL; K shows boxed region in inset (dmNes+RG and dVL regions). Red arrowheads point to dmNes+RG; green arrowheads point to delaminating dVL cells; blue arrowheads point to cells ventral to these. Representative images I–VI are shown at high magnification in panels. Green asterisks point to nucleus about to delaminate (double asterisk) or just delaminated (single asterisk). Red asterisks in panels I and II point to secretory vesicles. In vVL regions, the uniform arrangement of nuclei shows that each adhesion complex lies on either side of a cell. (L) Quantitative analysis showing length of electron-dense junctions along a single side of the VL: junction length increases dorsally and ventrally, from delaminating cells. Scale bars: A–J, 100 μm. aPKC, atypical protein kinase C; CRB2, Crumbs2; dmNes+RG, dorsal midline Nestin[(+)] radial glia; dVL, dorsal ventricular layer; EM, electron microscopy; PAR3, polarity protein PAR3; VL, ventricular layer; vVL, ventral ventricular layer; ZO-1, Zona occludens 1.

4E″ and S5D Fig), and then continuous labelling at E17 (Fig 4J). Together, these data suggest that apical adhesion/tight junction proteins are retained on dmNes+RG throughout collapse but are reduced on dVL progenitor cells; in particular, they are absent (Zo-1, aPKC, PAR3) or markedly reduced (CRB2) from the dVL cells that are delaminating.

To explore this further, we examined adhesion complexes after EM imaging (Fig 4K). dmNes+RG endfeet have long, tight adhesion complexes (Fig 4K red arrowheads), whose characteristic profiles (electron-dense; extending from the apical surface on adjacent cell membranes) can be detected at high magnification (Fig 4K panels I, II). By contrast, adhesion complexes can barely be detected on VL progenitor cells that are about to delaminate; only electron-lucent, short adhesion complexes can be detected (Fig 4K green arrowheads; panel IV). Immediately dorsal to these, a long adhesion complex extends parallel to the VL (panel III), potentially indicating a remodelling cell. In VL progenitor cells that lie ventral to delaminating cells, adhesion complexes lengthen and become increasingly electron-dense (Fig 4K blue arrowheads and panels V, VI). Quantitative measurements of the length of adhesion complexes reveal a gradual decrease in length in dVL cells that are about to delaminate, then a gradual increase towards dmNes+RG (Fig 4L).

In summary, throughout the collapse window, dmNes+RG form long apical adhesion complexes and tight junctions. By contrast, dVL progenitors that are about to delaminate show

markedly reduced apical adhesion complexes and tight junctions. Dorsal and ventral to these, apical adhesion complexes gradually increase in length. Together these analyses show that a reduction in apical adhesion/tight junction complexes prefigures or accompanies dVL cell delamination.

## dmNes⁺RG secrete a factor that promotes VL progenitor delamination

dmNes⁺RG have previously been implicated in dorsal collapse; studies in zebrafish have suggested that dmNes⁺RG cells are tethered to VL cells via a F-actin cytoskeleton belt and that tethering is required for dorsal collapse [45]. However, our EM imaging shows that dmNes⁺RG are rich in vesicles that appear to be fusing with the adjacent lumen (Fig 4K-I and 4K-II asterisks; see also [28]), suggesting that these cells could be secreting a factor involved in dVL cell delamination.

To test this idea, we established an in vivo assay, using the early chicken embryonic neural tube to assay dmNes⁺RG activity. Mouse E15 dmNes⁺RG cells or control VL cells (S6 Fig) were transplanted into the dorsal lumen of Hamburger-Hamilton (HH) stage 10 chick embryos, a stage when the neural tube is composed of pseudostratified neuroepithelial cells, and embryos were developed 20–24 hours, to HH stages 16–18 (Fig 5A schematic; *n* = 14 embryos each). In embryos transplanted with control lateral VL cells, the neural tube appeared normal: Zo-1 and aPKC were detected apically on VL progenitor cells (Fig 5C and 5D), the distribution of Pax6⁽⁺⁾ and Nkx6.1⁽⁺⁾ progenitors appeared normal, the basement membrane was intact (Fig 5E and 5F), and the secreted glycoprotein Sonic hedgehog (Shh) was detected on the floor plate (Fig 5G). By contrast, dmNes⁺RG cells had a marked effect on neuroepithelial cells. At 20 hours post-graft, Zo-1 and aPKC were down-regulated (Fig 5H and 5I). By 24 hours, Nkx6.1⁽⁺⁾ progenitor cells appeared disorganised (Fig 5K green arrowhead), the basement membrane showed breaks (Fig 5J and 5K white arrowheads), and some Pax6⁽⁺⁾ progenitor cells seemed to have delaminated from the neuroepithelium (Fig 5J green arrowhead). Ectopic clumps of Shh⁽⁺⁾ cells were similarly detected outside of the neural tube (Fig 5L arrowheads); in some cases, these appeared to be dissociating from the endogenous floor plate (S7A–S7B′ Fig). Therefore, dmNes⁺RG appear to provoke a loss of polarity, cohesion, and organisation of pseudostratified neuroepithelial cells. We next asked if other tissues could mimic these effects. Nestin⁽⁺⁾ cells in dorsal regions of the subventricular layer (SVL) of the lateral ventricle showed similar activity to dmNes⁺RG cells (S7C–S7G Fig). By contrast, other regions of the central nervous system (CNS) failed to mimic dmNes⁺RG cells (S5 Table).

The ability of transplanted dmNes⁺RG (and SVL cells) to affect progenitor cells at a distance indicated that these cells may secrete a factor that leads to down-regulation of apical polarity and tight junction proteins. Furthermore, this factor appears to initiate events that lead to loss of cohesion and organisation and may even promote neuroepithelial delamination.

## *Crb2S* is expressed in dmNes⁺RG and CRB2S mimics dmNes⁺RG activity

We had noted that during the period of dorsal collapse, CRB2 is not restricted to the apical endfeet of dmNes⁺RG but shows diffuse, punctate labelling in the endfeet (Fig 4C′ and 4C″ open arrowheads and Fig 5B and 5B′); similarly, punctate expression of CRB2 is detected in the Nestin⁽⁺⁾ SVL cells of the lateral ventricle (S7D and S7D′ Fig). Studies in humans suggest the existence of an alternatively spliced isoform of CRB2 that lacks the transmembrane domain and is putatively secreted [58], and work in *Xenopus* has shown a secreted variant of CRB2 (termed Xer1) in the early neural plate [59,60]. This prompted us to investigate whether a secreted isoform of CRB2 can be detected in mouse, and whether this protein can phenocopy the effects of dmNes⁺RG cell transplantation.

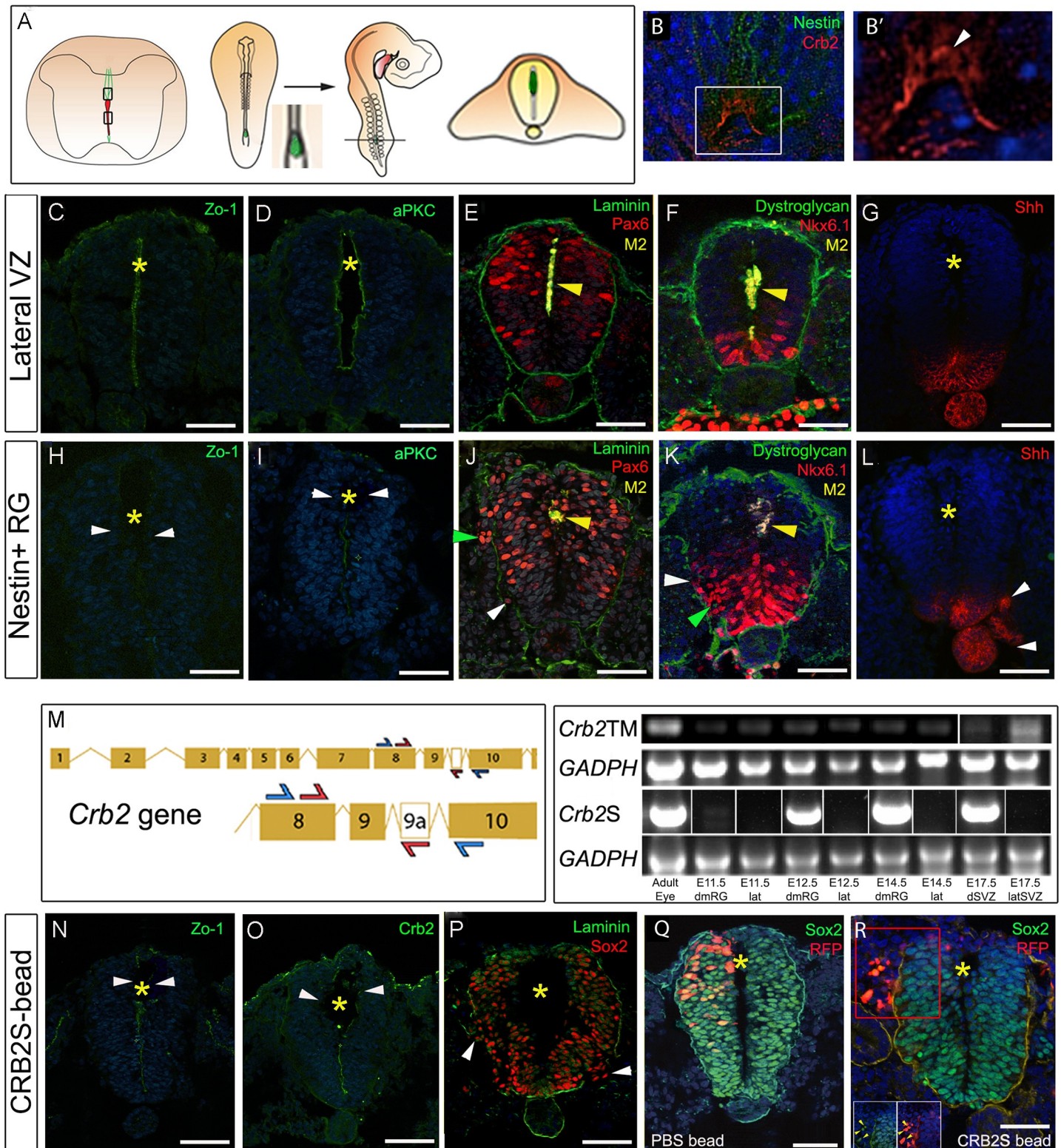

**Fig 5. CRB2S from dmNes⁺RG promotes loss of polarity and delamination.** (A) Schematic showing the grafting procedure: Mouse dmNes⁺RG or VL cells were dissected and grafted to Hamburger-Hamilton (HH) stage 10 chick embryos. After 20–24 hours of incubation (to HH stages 16–18), chicks were sectioned in the operated region, defined on the basis of the presence of or proximity to mouse tissue (all sections lie within 60 μm of mouse cells, analysed by α-M2 antibody; where tissue is present in section, it is marked by a yellow arrowhead; where not present, yellow asterisks show the position in a nearby section). (B-B′) Transverse sections of E16 mouse spinal cord double-labelled to detect Nestin and CRB2. Arrowhead in B′ points to non-apical CRB2 expression. (C-G) Transverse sections of HH stage 16–18

chick embryos after grafting control tissues, immunolabelled as shown. (C,D) Serial adjacent sections after a 20-hour incubation. Zo-1 and aPKC are detected on the apical side of neuroepithelial progenitor cells. (E-G) Serial adjacent sections after a 24-hour incubation. (H-L) Transverse sections of HH stage 16–18 chick embryos after grafting dmNes⁺RG. (H,I) Serial adjacent sections after a 20-hour incubation: little/no expression of Zo-1 and aPKC is detected (arrowheads). (J-L) Serial adjacent sections after a 24-hour incubation; ectopic Pax6$^{(+)}$ and Shh$^{(+)}$ cells are detected outside of the neural tube, the basement membrane shows breaks, and Nkx6.1$^{(+)}$ cells are disorganised (arrowheads). Underlying data can be found in S5 Table. (M) Left-hand panel: Schematic showing primer strategy against *Crb2* or *Crb2S*. CRB2S has an additional exon alternatively spliced into the transcript. Blue arrows indicate a first round of PCR amplifying a band between exon 8 and 10 present in both *Crb2* cDNA and *Crb2S* cDNA. Nested primers are indicated by red primers, which amplify between exon 8 and exon 9a, present in only *Crb2S*. Right-hand panel: Amplification (using the first round and nested primers) of adult eye, E11.5, E12.5, and E14.5 dmNes⁺RG (dmRG); E11.5, E12.5, and E14.5 lateral VL (lat); and dorsal or lateral E17.5 SVZ samples. A GADPH control is provided for both sets of primers in all samples. *Crb2S* lanes show representative images from 4 biological samples: see S7 Fig for full gel. (N-P) Transplanted CRB2S protein-soaked bead grafted to HH stage 10 chick embryo analysed after 24 hours. Zo-1 and Crb2 expression are reduced in the vicinity of the bead (N,O arrowheads). Sox2$^{(+)}$ neural progenitors are disorganised; some are detected outside the disrupted basement membrane (P, arrowheads). (Q) Control RFP electroporated and PBS-soaked bead-grafted HH stage 10 chick embryo after 24 hours shows normal expression of Sox2 throughout neural tube and normal basement membrane. (R) RFP electroporated and CRB2S-soaked bead-grafted HH stage 10 chick embryo after 24 hours shows ectopic expression of Sox2 outside neural tube and disrupted basement membrane. Red box indicates inset below. Yellow arrows indicate Sox2$^{(+)}$ RFP$^{(+)}$ cells outside the neural tube. Yellow asterisks in N-R shows position of bead (displaced on sectioning). Scale bars: C-L, N-R, 50 μm; B-B′, 10 μm. α-M2, anti-mouse astrocyte-surface antigen; aPKC, atypical protein kinase C; CRB2, Crumbs2; CRB2S, secreted Crumb2; dmNes+RG, dorsal midline Nestin$^{(+)}$ radial glia; GADPH, glyceraldehyde 3-phosphate dehydrogenase; Nkx6.1, NK6 homeobox 1; Pax6, paired-box 6; RFP, red fluorescent protein; RG, radial glia; Shh, Sonic hedgehog; Sox2, Sry-related HMG-box 2; SVZ, subventricular layer; VL, ventricular layer; VZ, ventricular zone; Zo-1, Zona occludens 1; $^{(+)}$, expressing.

Bioinformatic analysis of the mouse *Crb2* locus predicts different splice variants. One encodes the transmembrane variant of CRB2 ([CRB2TM]. In a second, however, alternative exon splicing between exon9 and exon10 introduces a premature stop codon before the transmembrane domain, predicting that this splice variant (which we term *Crb2+9A*, or *Crb2S*) may encode the putatively secreted protein. Stable clonal HEK293 cell lines constitutively expressing *Crb2+9A* secreted a CRB2-protein (termed CRB2S) into the media (S8 Fig and see Materials and methods). We therefore used a nested PCR approach (Fig 5M) to determine whether the mRNA encoding the secreted isoform can be specifically detected in dmNes⁺RG cells. dmNes⁺RG, SVL cells, and control lateral VL cells were compared to the adult eye, a tissue that expresses high levels of CRB2TM [61,62]. As predicted from immunohistochemical analyses, RNA encoding full-length CRB2TM was detected in both dmNes⁺RG cells and lateral VL cells. However, the RNA encoding the secreted isoform, CRB2S, was detected only in dmNes⁺RG and not in lateral VL cells (Fig 5M and S9 Fig).

Beads soaked in purified CRB2S were then implanted into the dorsal lumen of HH stage 10 chick embryos (*n* = 6). Transplants of CRB2S-soaked beads phenocopied transplants of dmNes⁺RG cells. Thus, the neural tubes of host embryos showed a reduction of apical polarity and junctional proteins, most obviously in the vicinity of the bead, disruption and disorganisation of the neuroepithelium, breaks in the basement membrane (Fig 5N–5P), and the ectopic appearance of neuroepithelial cells outside of the neural tube (Fig 5P and S10 Fig). PBS soaked beads caused no such effects (S10 Fig).

The appearance of ectopic progenitor and floor plate cells outside of the neural tube could arise, either through the delamination of neuroepithelial cells, or due to a fate change in cells outside of the neural tube. To distinguish between these, we combined RFP electroporation and bead implantation. When RFP was electroporated into the dorsal neural tube (avoiding the neural crest) prior to transplantation of a PBS bead, Sox2$^{(+)}$ RFP$^{(+)}$ cells were confined to the neural tube (Fig 5Q). By contrast, when similar cells were targeted, then a CRB2S bead implanted, Sox2$^{(+)}$ RFP$^{(+)}$ cells were detected outside of the neural tube (Fig 5R). Thus, in the presence of CRB2S, cells delaminate from the neuroepithelium. In summary, a secreted variant of CRB2, CRB2S, is specifically expressed in dmNes⁺RG cells, and its premature mislocalisation leads to loss of apical polarity proteins in neuroepithelial cells and their ability to delaminate from the neuroepithelium.

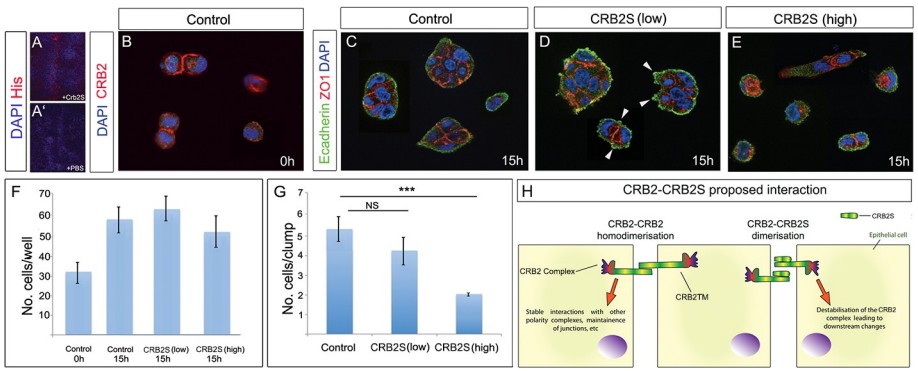

**Fig 6. CRB2S reduces cohesion/adhesion in MDCK cells.** (A,A′) Transverse sections of E15.5 spinal cord incubated with His-tagged CRB2S (A) or PBS (A′). Anti-His antibody detects CRB2S at the apical side of VL cells. (B) MDCK cells seeded at low density express CRB2 at junctions between doublets. (C-E) MDCK cells seeded at low density, then cultured an additional 15 hours in control medium (C), low concentration of CRB2S (D), or high concentration of CRB2S (E). Arrowheads in (D) point to blebs. (F,G) Quantitative analyses show that CRB2S does not affect the total cell number but significantly reduces the number of cells/clumps. Underlying data can be found in S7 and S8 Tables. (H) Model for effect of CRB2S. In the absence of CRB2S, CRB2TM binds homophilically [63] to promote stable interactions with components of the apical polarity/junctional complex and maintain neuroepithelial junctions. CRB2S competes homophilically to bind to CRB2. CRB2-CRB2S dimerisation destablilises the CRB2 complex, leading to downstream neuroepithelial destabilisation and delamination. CRB2, Crumbs2; CRB2S, secreted Crumbs2; CRB2TM, transmembrane CRB2; MDCK, Madin-Darby kidney cells; NS, non-significant; VL, ventricular layer; ZO-1, Zona occludens 1.

## CRB2S reduces cohesion/adhesion in MDCK cells

In zebrafish, CRB2 homologues can bind homophilically to mediate cell–cell adhesion in epithelial cells [63]. This raises the possibility that CRB2S can similarly associate with CRB2. To begin to test this, we incubated spinal cord sections with His-tagged CRB2S or control PBS, then analysed with an anti-His antibody. Punctate labeling was detected on the apical side of VL cells incubated with His-tagged CRB2S (Fig 6A) but not with control medium (Fig 6A′).

To test the effect of CRB2S on epithelial cells, we performed an acute assay on MDCK cells (a polarised columnar epithelial cell line [64,65]). In MDCK cells seeded at high density, the tight junction protein ZO-1 is apically located in 77% of cells cultured in control medium (S11A Fig). By contrast, apically localised ZO-1 is detected in only 23% of cells cultured with high concentrations of CRB2S (S11B Fig). We next examined the effect of CRB2S on MDCK cells seeded at low density. DAPI-labelling revealed the presence of single cells and doublets that express CRB2 (Fig 6B). Exposure of cells to control medium or to CRB2S for 15 hours suggested that CRB2S did not affect proliferation or apoptosis of MDCK cells: the total numbers of cells in each condition was not significantly different, and in each condition, a similar increase in total cell number was detected over the 15-hour culture period (Fig 6F). However, CRB2S had a negative effect on polarity and cohesion/adhesion. In control medium, the majority of cells were detected in small clumps (Fig 6C and 6G) and expressed ZO-1 and E-cadherin in a uniform manner (Fig 6C). After exposure to low concentrations of CRB2S, similar-sized clumps of cells were detected (Fig 6D and 6G). However, cellular E-cadherin-expressing blebs were apparent (Fig 6D arrowheads) and ZO-1 distribution was uneven and sometimes punctate (Fig 6D and S11C Fig). Furthermore, after exposure to higher concentrations of CRB2S, cells were largely detected as doublets (and sometimes singlets), with almost no larger clumps detected (Fig 6E and 6G). As with low CRB2S, ZO-1 showed unusual cytoplasmic expression, and E-cadherin was no longer uniformly detected in cells (Fig 6E). These observations suggest that CRB2S can reduce polarity and cohesion/adhesion in

CRB2-expressing cells. Together, these analyses suggest a model in which a direct interaction of CRB2S and CRB2TM destabilises CRB2TM-CRB2TM interactions, leading to downstream changes, including loss of polarity and decreased cohesion/adhesion (Fig 6H; see Discussion).

## *Crb2* and CRB2S are required for dorsal collapse

This model predicts that *Crb2* is essential for cell delamination and dorsal collapse but that this function occurs through an interaction of CRB2TM (required on dVL cells) and CRB2S (from dmNes+RG cells). To test this, we first deleted *Crb2TM* from neuroepithelial VL progenitors by crossing a transgenic *Crb2* floxed (*Crb2^{F/F}*) mouse (a construct designed to remove full-length *Crb2TM* but not *Crb2S* [41,66] with *Nestin-Cre^{+/−}* mice [67] to obtain *CRB2^{F/F}/Nestin-Cre^{+/−}*) animals (termed CRB2 Nestin-Cre hereafter). Embryos were compared to *Crb2^{F/F}* embryos at E17 (*n* = 4 embryos each). In control (*Crb2^{F/F}*) embryos, collapse occurred as normal. DAPI labelling showed the lumen was reduced to a similar extent to that seen in wild-type mice (compare Fig 7A and Fig 1A) and dmNes+RG cells elongated to a similar extent to that in wild-type embryos (compare Fig 7B and S2 Fig); CRB2 itself, aPKC, and PAR3 were detected on the apical side of VL cells (Fig 7C–7E) and PAX6^{(+)} and NKX6.1^{(+)} progenitors were located as in wild-type embryos (Fig 7F and 7G). By contrast, collapse failed to proceed

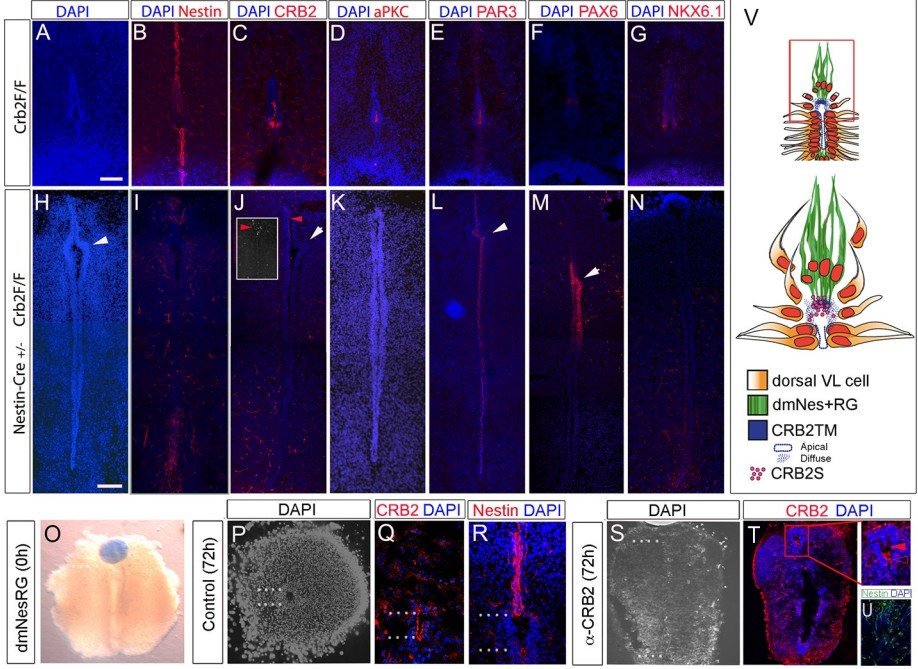

**Fig 7. CRB2TM and CRB2S are required for dorsal collapse.** (A-N) Transverse serial adjacent sections through E17 spinal cord in *CRB2^{F/F}* control embryos (A-G) or *CRB2^{F/F}/Nestin-Cre^{+/−}* embryos (H-N) after immunolabelling. Knockout of CRB2TM prevents collapse; a long lumen is detected, with unusual kinks (white arrowheads). Analysis of *CRB2^{F/F}/Nestin-Cre^{+/−}* with α-CRB2 Ab (J) reveals that little/no CRB2 is detected at the apical surface of VL cells, but punctate labelling is still detected dorsally. Inset shows a section from a second embryo, showing punctate labelling in dmNes+RG. (O-U) Whole-mount view (O) or serial adjacent sections (P-U) through spinal cord slices, cultured with PBS- or α-CRB2 Ab–soaked bead. White dots indicate length of lumen. In (T) CRB2 is detected at the apical surface of VL cells, including those at the dorsal pole, but no diffuse labelling is detected in dmNes+RG (arrowhead in boxed region). The view shown in (U) is that of the boxed region in (T). (V) Schematic shows model: CRB2S is secreted from the endfeet of dmNes+RG and binds to adjacent apical cells, resulting in their down-regulation of apical/junctional proteins, altered polarity, delamination, and migration along dmNes+RG scaffold. Scale bars: 50 µm. Ab, antibody; aPKC, atypical protein kinase C; CRB2S, secreted CRB2; CRB2TM, transmembrane CRB2; dmNes+RG, dorsal midline Nestin^{(+)} radial glia; PAR3, polarity protein PAR3; Pax6, paired-box 6; VL, ventricular layer.

 

normally in CRB2 Nestin-Cre embryos. DAPI labelling showed the dorsoventral extent of the lumen to be much greater than that detected in controls, consistent with a dorsal collapse failure (Fig 7H), although unusual kinks were detected in dorsal regions (Fig 7H, 7J, 7L and 7M arrowheads). Nestin immunolabelling showed that the ventral and dorsal RG failed to elongate (Fig 7I). Analysis with anti-CRB2 antibody confirmed a reduced expression of CRB2 at the apical surface of VL cells (Fig 7J), although punctate CRB2S persisted in dorsal regions (Fig 7J and 7J inset). Minimal expression of aPKC was detected (Fig 7K), but PAR3 was expressed at the apical side of VL cells (Fig 7L). PAX6$^{(+)}$ progenitors were retained in dorsal parts of the VL (Fig 7M), and in contrast to wild-type and $CRB2^{F/F}$ embryos, no Nkx6.1 expression could be detected on vVL cells (Fig 7N). Together, these data suggest that CRB2TM is required for dorsal collapse.

We next blocked CRB2S specifically from dmNes$^+$RG by implanting a bead soaked in α-CRB2 antibody adjacent to these cells in E13 mouse slice cultures (Fig 7O). After culture to an equivalent of E16, dmNes$^+$RG exposed to a control bead showed a collapse similar to that detected in vivo (Fig 7P). Analysis of transverse sections revealed apical CRB2 in remaining VL cells (Fig 7Q) and revealed elongated dmNes$^+$RG (Fig 7R). By contrast, collapse failed to occur when dmNes$^+$RG were exposed to an α-CRB2 antibody-soaked bead (Fig 7S). The diffuse labelling characteristic of CRB2S (Fig 4C and 4C′ and Fig 5B and 5B′) could not be detected in cells at the dorsal pole (Fig 7T red arrowhead), although apical CRB2 could still be detected within VL cells (Fig 7T white arrowhead), and dmNes$^+$RG failed to extend (Fig 7U). Similar results were observed when CRB2 was deleted from dmNes$^+$RG after targeting a $shCrb2$ construct [68] in E13 mouse slice cultures (S12 Fig).

Together, these results show that CRB2 is required both in VL cells and dmNes$^+$RG for dorsal collapse, and that the CRB2S variant mediates the activity of the dmNes$^+$RG.

## Discussion

Here, we present evidence for a novel role for the apical protein CRB2 in a local delamination and ratcheting mechanism that remodels the VL of the vertebrate spinal cord into the EL. Similar to recent studies on mouse gastrulation [41,69,70], our data show that $Cbr2$ is required for the removal of cells within an epithelium. Such a role for $Crb2$ is surprising because, classically, $Crb2$ and drosophila $crumbs$ are required for polarity and the cohesion and maintenance of epithelia [36,71,72]. Our studies show that in the spinal cord, the ability of CRB2 to effect cell delamination is mediated by a secreted variant, CRB2S. We propose that a CRB2TM–CRB2S interaction causes changes in cell polarity and cell cohesion that ultimately effect cell delamination and mediate dorsal collapse. Furthermore, the ratcheting interaction of CRB2S dmNes +RG cells and CRB2TM VL cell suggests a mechanism that enables EL and central canal formation via attrition of VL cells without loss of overall VL/EL integrity.

CRB2 is detected in a dynamic pattern in cells that line the spinal cord lumen. dmNes$^+$RG that tether the lumen to the pial surface maintain high levels of CRB2 in their apical endfeet throughout collapse. By contrast, although CRB2 is detected on the apical sides of all VL cells prior to and following collapse, it is eliminated from the apical side of dVL cells that immediately neighbour dmNes$^+$RG during collapse. Full-length transmembrane $Crb2$ is expressed in all VL cells, but a splice variant that lacks the transmembrane domain and is predicted to be secreted [58,62,73] is specifically expressed in dmNes$^+$RG. Four lines of evidence suggest that CRB2S secreted from dmNes$^+$RG acts on immediately adjacent dVL cells to disrupt apicobasal polarity and effect their delamination from the VL. First, the CRB2 splice variant can be secreted, and premature exposure of chick neuroepithelial VL cells to CRB2S leads them to lose apical aPKC and even to delaminate. Second, exposure of CRB2-transfected MDCK cells

 

to CRB2S disrupts apicobasal polarity, as assessed through the basal protein, E-cadherin, and the apical tight junction protein, ZO-1 (similarly reduced in chick neuroepithelial cells after exposure to CRB2S), and reduces their cohesion/adhesion. Third, the phenotype elicited by CRB2S is recapitulated by dmNes+RG: premature and ectopic exposure of chick neuroepithelial VL cells to dmNes+RG, but not other spinal cord VL cells, leads to the loss of Zo-1, aPKC, and delamination of progenitor cells. Fourth, targeted blockade of CRB2S in dmNes+RG prevents dorsal collapse. It remains to be proven that CRB2S is secreted in vivo, as appears to occur in *Xenopus* [59,60], but these observations, together with the secretory nature of dmNes+RG and their specific expression of the *Crb2* splice variant, provide strong evidence for this idea.

Our demonstration that CRB2S associates with CRB2 on the apical side of VL cells suggests a mechanism for how dVL delamination may be initiated. In flies and vertebrates, full-length CRB2 homologues normally mediate cell adhesion through homophilic interactions at opposing cell membranes, mediated by the extracellular domain. Homophilic interactions stabilise Crb/CRB2 apically, and maintain epithelial organisation and integrity [36,43,44,63,68,74–76]. The full mechanisms through which this is achieved remain elusive but likely involve the ability of CRB2 to interact with the PAR complex (Cdc42-Par6-aPKC-PS980-Baz), which maintains apicobasal polarity and adherens junction assembly and positioning [36,63,75–77]. In flies, the Crb transmembrane domain retains Cdc42-Par6-aPKC at the apical part of the cell, enabling PS980-Baz (Bazooka) to accumulate at the lateral part of the cell and recruit adherens junction material, including E-cadherin, a key intercellular adhesion factor [36,77]. Our studies suggest that the association of CRB2S with CRB2 within the vertebrate spinal cord disrupts this pathway. In vivo, VL cells closest to dmNes+RG down-regulate the apical polarity proteins CRB2, aPKC, PAR3, and the tight junction protein ZO-1. In experiments, exposure of cells to CRB2S results in the loss of aPKC, the mediator of PAR complex signalling [36,78,79], and disrupts E-cadherin localisation. Together, our findings suggest a model (Fig 6H and Fig 7V) in which all VL cells, including dmNes+RG cells, express CRB2, which supports apicobasal polarity and cohesion; however, dmNes+RG also secrete CRB2S, which acts non cell-autonomously and locally to compete away CRB2 in neighbouring cells, leading to loss of apicobasal polarity.

Our studies suggest that the CRB2S-CRB2TM–mediated polarity changes exert specific downstream effects to enable delamination, which are not triggered simply through genetic loss of full-length CRB2 (Fig 7J). Our imaging studies show that intimate interactions between dmNes+RG and dVL cells lead to stepwise dVL delamination; live imaging shows that as each dVL cell delaminates, the next dVL cell ratchets up, the dmNes+RG endfoot ratchets down, and the process repeats. Additionally, we detect a remodelling of cells that are delaminating, including a dorsal 'reaching' (S3 Movie), a dorsal reorientation of nuclei (Fig 2), and an unusually long adherens complex, parallel to the VL (Fig 4K panel III). A prosaic interpretation is that these events are triggered by the CRB2S-CRB2TM interaction and enable a specific delamination mechanism. Our observations that PAR3 (the vertebrate homologue of Baz, which recruits adherens junction material [36,77], is down-regulated during dorsal collapse (Fig 4D–4D″) but not after genetic loss of *Crb2TM* (Fig 7L) raises the possibility that PAR3 down-regulation is needed for the specific delamination detected in dorsal collapse.

Live imaging reveals that as each dVL cell ratchets up, the dmNes+RG endfoot ratchets down. Previous in vivo studies in zebrafish have suggested that dmNes+RG cells are tethered to VL cells via a F-actin cytoskeleton belt, that tethering is required for dorsal collapse and depends on the activity of Rho-associated protein kinase (Rok) [45], so potentially, an actin cable ratchets dmNes+RG end-feet to the next VL cell and tensions the diminishing VL. A second possibility not exclusive to this is that the reaching of the dVL cell onto the dmNes+RG promotes the lengthening of the latter.

Our findings do not address how VL progenitor cell delamination initiates or terminates. Little CRB2 is detected in dmNes⁺RG at E13.5 (S3B Fig) prior to dorsal collapse, and relatively little *Crb2S*-encoding mRNA is detected at E11.5 (Fig 5M), but we cannot exclude an earlier role for CRB2S, for instance, in neural crest–neural plate boundary formation, similar to that observed in *Xenopus* [59,60]. Furthermore, the punctate, non-apical CRB2 that we detect at E15.5 (Fig 5B and 5B′), potentially CRB2S, is not obviously detected at the end of dorsal collapse (S3E and S3F Fig). Another possibility, not exclusive, is that vVL cells are, or become, refractory to the action of CRB2S. Future studies are needed to understand this question, and that of why dmNes⁺RG are themselves refractory to the action of CRB2S.

The mechanism that we uncover in these studies is likely to be one of multiple steps that contribute to spinal cord VL attrition and eventual formation of the adult central canal. The looser arrangement of dVL cells and their lack of expression of ZO-1 indicate that the attrition process we describe here is only one of multiple integrated mechanisms. Previous studies, for instance, indicate a role for declining proliferation [1] and Wnt signalling [20,28]. Moreover, a parallel study in one of our laboratories reveals that central canal formation proceeds through a combination of cell rearrangements at each pole: dorsally, dorsal collapse, and ventrally, a dissociation of a subset of floor plate cells, accompanied by changes in the activity of critical ventral and dorsal patterning signals: a gradual decline in ventral sonic hedgehog activity and an expansion of dorsal bone morphogenetic protein signalling [1]. Our data here provide evidence that CRB2 is involved in these additional events: in mice that lack CRB2, cell fate specification mediated by patterning signals is dysregulated; no expression of the homeobox transcription factor, NKX6.1, is detected on aberrantly retained VL progenitors. Further studies are needed to determine whether CRB2 governs NKX6.1 through its ability to interact with other homeobox genes [80] or via an effect on SHH-BMP signalling.

In conclusion, our studies reveal a novel mechanism of action of CRB2, in which the action of a secreted variant, CRB2S, deriving from dmNes⁺RG, mediates loss of apicobasal polarity and local delamination as part of a mechanism that establishes the central canal. The finding that CRB2S is expressed elsewhere in the CNS suggests it may operate more widely to promote local delamination: future studies are needed to establish whether its expression in SVL cells of the lateral ventricle, a recognised stem cell niche in the brain, promotes delamination associated with neuronal differentiation, and whether its expression in the eye is involved in dynamism of the retinal neuroepithelium, where loss of CRB2 leads to retinal degeneration [41,66,68,81,82]. More generally, our findings demonstrate how early patterning centres (dmNes⁺RG derive from roof plate cells) are maintained through life to support remodelling and maintenance. Finally our studies add to the evidence that, similar to *Drosophila* epithelial sheets, the vertebrate neuroepithelium is modelled by dynamic local cell–cell interactions, and reveal a cell non-autonomous action for CRB2S in neuroepithelial remodelling.

## Materials and methods

### Ethics statement

All procedures concerning transgenic mice were performed with the permission of the animal experimentation committee (DEC) of the Royal Netherlands Academy of Arts and Sciences (KNAW) (permit and approval number NIN06–46), and all experiments for wild-type mice were approved by the University of Sheffield Animal Welfare and Ethical Review Body (AWERB) and conformed to the United Kingdom Home Office ethical guidelines.

## Mice

C57BL/6J or CD1 mice were used to obtain wild-type mouse embryos. Timed mating was used to obtain embryos at the appropriate stages. Pregnant mice were anaesthetised to unconsciousness through isoflurane inhalation (Abbot Laboratories, Sittingbourne, UK) before cervical dislocation. Embryos were transferred into Leibovitz's 15 (L-15 Gibco, Thermo Fisher Scientific, Grand Island, NY, and Paisley, Scotland) before decapitation, then further dissection.

## Generation of transgenic mice

To generate *CRB2* Nestin-Cre cKO (*Crb2$^{F/F}$/Nestin-Cre$^{+/-}$*) mouse embryos, homozygote *CRB2$^{F/F}$* mice [41] were crossed at the Netherlands Institute for Neurosciences with double heterozygote *Crb2$^{F/+}$/Nestin-Cre$^{+/-}$* [67]. Nestin-Cre expressing cells specifically delete *Crb2* encoding exons 10–13 [66]. Note *Crb2* Nestin-Cre cKO mice die shortly after birth. For the analysis of the mutant mouse models, embryos were genotyped and sent to the UK in 30% sucrose solution. Three control and three conditional knockout embryos were used for marker analysis.

## Immunohistochemistry

Tissues were fixed in 4% paraformaldehyde for 2 hours, transferred to 30% sucrose overnight, then cryosectioned at 15 μm and incubated with primary antibodies: anti-Nestin (1:300, Abcam plc, Cambridge, UK), anti-NKX6.1/Nkx6.1 (1:50, Developmental Studies Hybridoma Bank [DSHB], IA), anti-SOX1 (1:300, Cell Signalling Technologies, London, UK), anti-SOX2/Sox2 (1:1,000, Millipore, Watford, UK), anti-SOX3 (1:1,000, gift from T. Edlund), anti-ZO-1/Zo-1 (1:300, Zymed, Thermo Fisher Scientific, Runcorn, UK), anti-phalloidin (1:500, Thermo Fisher Scientific, Runcorn, UK), anti-aPKC (1:300, Santa-Cruz Biotechnology, Heidelberg, Germany), anti-PAR3 (1:200, Millipore, Watford, UK), anti-CRB2 (1:300, [83]), anti-dystroglycan MANDAG-2 (1:30, gift from S. Winder), anti-M2 (mouse astrocyte-surface antigen) (1:50, DSHB), anti-Transitin (1:50, DSHB), anti-PAX6/Pax6 (1:50, DSHB), and anti-pH3 (1:1,000, Millipore, Watford, UK). Alexa 488– and 594–conjugated secondary antibodies were used (1:500; Thermo Fisher Scientific/Molecular Probes, cat. nos. A11001, A11034, and A11005). Slides were mounted in Vectashield (Vector Laboratories, Romford) and analysed. For each antibody, three sections were analysed from 3 embryos at each stage. For analysis of laminin/dystroglycan, breaks in basement membrane were scored as regions >3 nuclei in length, to avoid counting small tears.

## RT-PCR

The *Crbs2S* RT-PCR reaction was performed on cDNA synthesised from tissues using SuperScript III First-Strand Synthesis System (Invitrogen, Thermo Fisher Scientific). *Crb2* mRNA was amplified. Primers were designed to amplify full-length mature *Crb2* and *Crbs2* (*Crb2F* TGTATGTGGGTGGGAGGTTC [Exon8; Tm 59.00]; Crb2R TAACGGGAAGTCGCCAAGT [Exon 10; Tm 59.0]). A second round of PCR was then performed, designed to amplify *Crb2S* specifically (*Nested F* CTACAACTCAACAGCATCC [Exon 8 Tm 59.2]; *Nested R* GCTTC GGTTGGTAGACTGCC [Exon 9a Tm 58.3]). A GAPDH loading control was run (*GAPDHF* AACGGGAAGCCCATCACC [Tm 59.7]; *GAPDHR* CAGCCTTGGCAGCACCAG [Tm 58.0]). The reactions were run on an agarose gel with the addition of ethidium bromide (Bio-Rad, Watford, UK), and bands of the appropriate size were excised using QiAquick Gel Extraction Kit (Qiagen, Manchester, UK) and sequenced in-house.

## Slice culture and live imaging

Freshly dissected E13 mouse thoracic/lumbar spinal cord or E7 chick spinal cords were electroporated with GFP-GPI, RFP-H2B [7], and pAAV-CMV-eGFP-shCrb2[68] plasmids at a concentration of 0.2–0.9 ug/uL. This region of the embryo was then mounted in 4% low melting point agarose (Sigma) and sectioned at 300 μm on a vibrating blade microtome (Leica VT1200 S). Slices were then embedded in collagen on glass-bottomed imaging dishes (World Precision Instruments, Hitchin) and incubated at 37˚C 50% $CO_2$ in Neurobasal medium (Gibco, Thermo Fisher Scientific) supplemented with Glutamax (Gibco, Thermo Fisher Scientific), B27 (Life Tech, Thermo Fisher Scientific), foetal calf serum (Sigma, Dorset, UK), and Gentamycin (Gibco, Thermo Fisher Scientific) for 24 hours before imaging. Live imaging was performed as described in [7].

## Cell culture

Confluent MDCK cells were trypsinised and resuspended. Cells were replated at low density. The cells were incubated for 6 hours and allowed to reattach. CRB2S or PBS was added to the media. Cells were fixed and analysed after a 15-hour incubation.

## Light microscopy and image analysis

Fluorescent images were taken on a Zeiss Apotome 2 microscope with Axiovision software (Zeiss) or, for high magnification images, on a Nikon Ti system running Nikon Elements AR software or a Deltavision RT system running SoftWorx. Mouse time-lapse images were taken on a Deltavision Core system enclosed in an environment chamber maintained at 37˚C and 5% $CO_2$. Images were acquired using a 40× 1.3 NA oil immersion objective (Olympus), led light source (Applied Precision), and a CoolSNAP HQ2 CCD camera (Photometrics). Images were deconvolved in Huygens Professional (Scientific Volume Imaging) and processed using Image-J (FIJI). Chick time-lapses were taken on a Zeiss Cell Observer system enclosed in a chamber maintained at 37˚C and 5% $CO_2$. Images were acquired using a 40× 1.2 NA silicone immersion objective (Carl Zeiss), LED light source (Carl Zeiss), and a Flash4 v2 sCMOS camera (Hamamatsu). Images were deconvolved and processed using the Zen Blue software (Carl Zeiss).

## Transmission EM

Specimens were fixed in 3% glutaldehyde/0.1 M sodium cacodylate buffer overnight, washed in buffer and dehydrated in ethanol, cleared in epoxypropane (EPP) and infiltrated in 50/50 araldite resin:EPP mixture on a rotor. This mixture was replaced twice over 8 hours with fresh araldite resin mixture before embedding and curing at 60˚C for 48–72 hours. Ultrathin sections, approximately 85 nm thick, were cut on a Leica UC 6 ultramicrotome onto 200 mesh copper grids, stained for 30 minutes with saturated aqueous uranyl acetate, followed by Reynold's lead citrate for 10 minutes. Sections were examined using a FEI Tecnai Transmission Electron Microscope at an accelerating voltage of 80 Kv. Electron micrographs were recorded using a Gatan Orius 1000 digital camera and Gatan Digital Micrograph software.

## In vivo manipulations

Chick embryos were staged using the Hamburger-Hamilton embryo staging chart, and the clear vitelline membrane was removed over the caudal neural tube into which the tissue/bead was to be transplanted. Freshly dissected embryonic mouse spinal cord was sliced into 400-μm sections on a tissue chopper (McIlwain) and the slices placed into ice-cold L-15. Tissue to be

transplanted was punched out with a pulled glass needle (1 mm × 0.78, Harvard Apparatus) and mouth pipette before being carefully placed into the most rostral part of the open neural tube. Sister punches were evaluated with anti-Nestin or anti-NKX6.1 antibodies to confirm that dmNes⁺RG or lateral/vVL cells could be isolated accurately and free from contaminating tissue (S6 Fig). Affi-gel beads (Bio-Rad) were soaked in protein or PBS control for 24 hours before transplantation into RFP electroporated/non-electroporated HH stage 10 chick embryos. Beads were carefully placed into the most rostral part of the open neural tube. The egg was sealed before incubation at 37˚C for 24 hours. Embryos were then dissected out and fixed in ice-cold 4% PFA in for 2 hours prior to sectioning and immunohistochemistry. Operated regions were identified through the presence of the mouse tissue (α-M2 antibody).

## Generation of stable cell lines and CRB2S protein purification and sequencing

HEK293 cells were transfected with either CRB2S cDNA or CRB2 signal peptide cDNA expression vectors. The cells were grown and expanded in selective conditions—medium + G418 (800 μg/mL, Sigma Aldrich, Dorset, UK). The expression level of protein of interest in the clones was determined using V5-tagged protein in the processed cell culture supernatant by western blotting. Three positive clones were expanded.

A HEK 293 stable cell line overexpressing *Crb2S* was used for obtaining purified protein. The transgenic *Crb2S* cell line was passed onto BioServ UK for scale-up of cells and immobilised metal ion affinity chromatography (IMAC). The cells were maintained in G418 selection antibiotic (800 μg/mL) throughout the culture period. The purified CRB2S protein (100 μg/ mL) was sequenced as described below, aliquoted, and stored at −80˚C. For protein sequencing, SDS gel electrophoresis was carried out as described below; care was taken to minimise external keratin contamination from the environment. All processing was carried out in a clean biosafety cabinet. The gel was fixed and stained with Coomassie Brilliant Blue (Sigma-Aldrich, Dorset, UK) as per the manufacturer's instructions, and the bands of interest were excised using a clean blade and stored at 4˚C in a sterile tube. LC-ESI-Mass spectrometry was carried out by a commercial company (Eurogentec) using an LC (nano-Ultimate 3000- Dionex)-ESIion trap (AMAZONE-Bruker) in positive mode.

## Western blotting

Western blotting was carried out using the NuPAGE gel system (Invitrogen, Thermo Fisher Scientific) according to the manufacturer's instructions. Cells were lysed in RIPA buffer (Sigma Aldrich, Dorset, UK) supplemented with protease inhibitor tablets (complete Mini, EDTA-free, Roche Products, Welwyn Garden City, UK) on ice. Total protein lysate (20 μg), as determined by Bradford assay, per sample was loaded on a 4%–12% gradient gel (Invitrogen, Thermo Fisher Scientific) under denaturing conditions. Membranes (Hybond-C Extra, Amersham Biosciences) were blocked in 5% semi-skimmed milk powder, Tween (0.1%) PBS, and primary antibodies (V5 Tag Abcam, chicken polyclonal 1:2,000, His Tag Cell signalling, rabbit polyclonal 1:1,000) were applied in blocking solution overnight at 4˚C. Secondary antibodies linked to horseradish peroxidase (all from Stratech Scientific, Ely, UK) were applied for 1 hour at a concentration of 1:1,000.

## Supporting information

**S1 Fig. dVL cells remodel in dorsal collapse.** All panels show high-magnification views of the VL in transverse sections at different time points. In (A-F), white bracket demarcates vVL. (A, A′) At E16, DAPI labelling reveals mediolaterally oriented vVL cell nuclei around a narrow

lumen, and diagonally oriented dVL cell nuclei around a wider lumen. (B, B′) At E17, dVL cell nuclei are dorsoventrally oriented. (C-F) SOXB1-immunolabelled cells at E15.5 (C,D), E16 (E), and E17 (F). Red bracket demarcates dVL; yellow arrowheads show dissociating floor plate cells; orange arrowheads show excluded SOXB1[(+)] cells in the dorsal midline. (G-J) Pax6 labelling at E14 (G), E15 (H), E16 (I), and E17 (J). PAX6[(+)] cells are excluded in dorsal collapse (inset, J). Scale bars: A-F, 50 μm; G-J, 100 μm. dVL, dorsal ventricular layer; PAX6, paired box 6; SoxB1, SRY-related HMG-box B1 transcription factors; VL, ventricular layer; vVL, ventral ventricular layer.
(TIF)

**S2 Fig. Nestin marks dmNes[+]RG.** Transverse sections through mouse embryonic spinal cord. (A) At E14 Nestin is detected on mediolateral radial glia. (B, C) By E15, strong Nestin labelling is detected on dorsal and ventral midline radial glial cells (white arrows) that project through the dorsal and ventral funiculi (yellow arrowheads) to the pial surface. Dotted lines demarcate lumen ends. dmNes[+]RG, dorsal midline Nestin[(+)] radial glia.
(TIF)

**S3 Fig. Cell behaviours during dorsal collapse.** (A) Sequential stills from time-lapse imaging (S2 Movie) after high-density electroporation of membrane-GFP histone-RFP into mouse spinal cord slice. Yellow arrowhead points to a dmNes[+]RG, whose position remains the same throughout the culture. Red arrowhead points to a nucleus that migrates dorsally. (B) Sequential stills from time-lapse imaging (S4 Movie) after low-density electroporation of membrane-GFP into chick spinal cord slice. (B′) Same images as in (B); cells colour-coded. dmNes[+]RG cell (red) elongates (9–12 hours) to contact a dVL cell (pink; contact at 9 hours); on the other side, a dVL cell (blue) ratchets up to the dmNes[+]RG (0–5 hours). dmNes[+]RG, dorsal midline Nestin[(+)] radial glia; dVL, dorsal ventricular layer; GFP, green fluorescent protein; RFP, red fluorescent protein.
(TIF)

**S4 Fig. Dorsal collapse in the chick spinal cord.** (A, A′–E, E′) Serial adjacent transverse sections through chick embryonic spinal cord between developmental stages E7 and E11. Expression of Sox2/Transitin (a Nestin-like protein) mirrors that of Sox2/Nestin in mouse embryonic spinal cord. As in mouse, dmTransitin[+]RG stretch from the lumen to the pia. Sox2 cells are found throughout the collapsing VL, as well as dorsal to the obliterated lumen, closely associated with Transitin[(+)] radial glial processes. dmTransitin[+]RG, dorsal midline Transitin-expressing radial glia; Sox2, SRY_related HMG-box 2; VL, ventricular layer.
(TIF)

**S5 Fig.** (A-E) Transverse sections through E13.5 (A-C), E17 (E), or E18 (F) mouse spinal cord, analysed by immunohistochemistry as shown. (D) Plots show intensity of labelling along the apical side of VL, from vVL (red) to dVL (blue) to dmNes+RG (dark red) in representative sections analysed at E15.5. dmNes+RG, dorsal midline Nestin[(+)] radial glia; dVL, dorsal ventricular layer; VL, ventricular layer; vVL, ventral ventricular layer.
(TIF)

**S6 Fig.** (A, B) Transverse sections, immunolabelled to show position of dmNes[+]RG (A) or NKX6.1[+] vVL cells (B). Circles indicate punched regions. (C-F) Accuracy of punches confirmed through immunolabelling. dmNes[+]RG express Nestin but not Nkx6.1 (C,D). vVL cells express NKX6.1 but not Nestin (E,F). dmNes[+]RG, dorsal midline Nestin[(+)] radial glia; NKX6.1, NK6 homeobox 1; vVL, ventral ventricular layer.
(TIF)

**S7 Fig. dmNes⁺RG and SVZ cells promote delamination.** (A-C) Transverse sections through HH st14 chick embryonic neural tube, 24 hours after transplantation with E15.5 dmNes⁺RG tissue. Shh⁽⁺⁾ floor plate cells appear to dissociate. (C) Schematic showing position of SVZ in mouse E17.5 telencephalon. (D) Dorsal SVZ cells co-express Nestin and CRB2; the latter appears non-apical. (D′) High-power view of boxed region. (E-G) Transverse sections through HH st14 chick embryonic neural tube, 24 hours after transplantation with E17.5 mouse SVZ tissue. (E) Shh is detected on cell clumps that appear to have dissociated from the floor plate. (F,G) Sox2 and Nkx6.1 progenitors are located ectopically outside the neural tube. CRB2, Crumbs2; dmNes⁺RG, dorsal midline Nestin⁽⁺⁾ radial glia; Nkx6.1, NK6 homeobox 1; Shh, Sonic hedgehog; Sox2, SRY-related HMG-box 2; SVZ, subventricular zone.
(TIF)

**S8 Fig. CRB2S is detected in the cell culture supernatant after exogenous overexpression in HEK293 cells.** (A) *Crb2S* cDNA and *Crb2S* signal peptide coding cDNA were cloned into pcDNA3.1 V5-His-Top expression vector. (B) Western blotting to detect the V5-tagged recombinant protein shows that CRB2S can be detected in the supernatant (S) and lysate (L) from cells transfected with the *Crb2S* expression vector. Cells were cultured in serum-reduced conditions for 72 hours before harvesting. GAPDH was used as a loading control. Apparent molecular weights are indicated on the left in B. CRB2S, secreted CRB2; GAPDH, glyceraldehyde 3-phosphate dehydrogenase.
(TIF)

**S9 Fig. Amplification using nested primers, of adult eye, E11.5, E12.5, and E14.5 dmNes +RG (dmRG); E11.5, E12.5, and E14.5 lateral VL (lat); and dorsal or lateral E17.5 SVZ samples.** dmNes⁺RG, dorsal midline Nestin⁽⁺⁾ radial glia; SVZ, subventricular zone; VL, ventricular layer.
(TIF)

**S10 Fig.** Transverse serial adjacent sections through HH st14 chick embryonic neural tubes, 24 hours after implantation of PBS-soaked (A-C) or CRB2S-soaked (D-F) beads. (A-C) PBS-soaked beads do not disrupt Pax6⁽⁺⁾ dorsal progenitors, Nkx6.1⁽⁺⁾ ventral progenitors, or Shh⁽⁺⁾ floor plate cells. (D-F) CRB2S-soaked beads caused delamination of neural tube progenitors: Pax6⁽⁺⁾ and Nkx6.1⁽⁺⁾ progenitors are mislocalised/mispatterned and detected outside of the neural tube (arrowheads). Shh expands dorsally and is detected on cell clumps that appear to have pinched off from the floor plate. Asterisk in (D) points to bead. Underlying data shown in S5 Table. CRB2S, secreted CRB2; Nkx6.1, NK6 homeobox 1; Pax6, paired-box 6; Shh, Sonic hedgehog; Sox2, SRY-related HMG-box 2.
(TIF)

**S11 Fig.** MDCK cells, cultured at high density in control medium (A) or a high concentration of CRB2S (B) or at low density with a low concentration of CRB2S (C), immunolabelled with Zo-1 and E-cadherin. XZ-plane views show a disruption in polarity in the presence of CRB2S. Underlying data shown in S6 Table. CRB2S, secreted CRB2; ZO-1, Zona occludens 1.
(TIF)

**S12 Fig. Mouse slice cultures: dmNes⁺RG targeted at E13 and cultured to E16 equivalent.** (A) At 0 hours, fast-green shows targeted electroporation to roof plate/dmNes⁺RG. (B-B″) After a 72-hour culture, slices targeted with a control GFP construct showed a 4-fold collapse, i.e., similar to that in vivo. Analysis of whole-mount slices showed GFP at the dorsal lumen (B), and analysis of sections revealed GFP in elongated dmNes⁺RG (B′,B″). (C-C″) By contrast, after targeting dmNes⁺RG with *shCrb2*, no collapse is detected (C) and no elongated

dmNes$^+$RG can be detected (C′,C″). dmNes$^+$RG, dorsal midline Nestin$^{(+)}$ radial glia; GFP, green fluorescent protein.
(TIF)

**S1 Table. Length (in μm) of the spinal cord VL at thoracic levels.** Three embryos were analysed at each stage; each row shows measurement from one 15-μm section. VL, ventricular layer.
(DOCX)

**S2 Table. Length (in μm) of the dVL and vVL at thoracic levels on consecutive days (E14–E17).** Three embryos were analysed from each stage; each row shows measurement from one 15-μm section. Unpaired Student $t$ test shows significant differences in length of dVL on each consecutive day but no significant difference in length of vVL on consecutive days. $^{***}p < 0.001$; $^{**}p = 0.0011$. dVL, dorsal ventricular layer; vVL, ventral ventricular layer.
(DOCX)

**S3 Table. Number of VL nuclei expressing SOX2, NKX6.1, or PAX6 on consecutive days; each row shows measurements from two 15-μm sections from one embryo.** Between E14 and E17, there is a large and similar proportional reduction of SOX2$^{(+)}$ and PAX6$^{(+)}$ nuclei (83.6% and 86.9%, respectively), but only a small reduction (23.5%) of NKX6.1$^{(+)}$ nuclei. Nkx6.1, NK6 homeobox 1; PAX6, paired-box 6; SOX2, SRY-related HMG-box 2; VL, ventricular layer.
(DOCX)

**S4 Table. Density of nuclei in dVL and vVL (nuclei/100 μm2).** Analyses based on measurements from 2 embryos, 2 sides. dVL, dorsal ventricular layer; vVL, ventral ventricular layer.
(DOCX)

**S5 Table. Transplantation of cells or beads to HH stage 10 chick embryo.** Number of embryos showing strong, subtle, or no phenotype after transplantation of dmNes$^+$RG, VL cells, ventral radial glia (RG), CRB2S-soaked beads, PBS-soaked beads, or SVZ cells. CRB2S, secreted CRB2; dmNes$^+$RG, dorsal midline Nestin$^{(+)}$ radial glia; SVZ, subventricular zone; VL, ventricular layer.
(DOCX)

**S6 Table. MDCK cells plated at high density and cultured in control medium or CRB2S.** After culture, cells were immunolabelled to detect ZO-1 and E-cadherin. Five random fields were selected ($n$ = 2 experiments) and XZ-plane views analysed for 20 cells per field. Cells were scored as polarised if ZO-1 was detected apically. Table shows number cells/field showing apical ZO-1; mean values and SEM shown in bottom line. CRB2S, secreted CRB2; ZO-1, Zona occludens 1.
(DOCX)

**S7 Table. MDCK cells plated at low density.** Five random wells were selected and cells counted at 0 hours or after a 15-hour culture in control medium, CRB2S (low concentration), and CRB2S (high concentration). Bottom line shows mean values and SD. CRB2S, secreted CRB2.
(DOCX)

**S8 Table. MDCK cell plated at low density.** Three random fields were selected (from 3 random wells) and a total of 15 clumps counted after a 15-hour culture in control medium, CRB2S (low concentration), and CRB2S (high concentration). Bottom line shows mean values

and SEM. CRB2S, secreted CRB2.
(DOCX)

**S1 Movie. Time-lapse imaging of mouse spinal cord slice culture after electroporating low numbers of dorsal cells with membrane-GFP histone-RFP.** Dorsal up. GFP, green fluorescent protein; RFP, red fluorescent protein.
(MP4)

**S2 Movie. Time-lapse imaging of mouse spinal cord slice culture after electroporating high numbers of dorsal cells with membrane-GFP histone-RFP.** Dorsal up. GFP, green fluorescent protein; RFP, red fluorescent protein.
(MOV)

**S3 Movie. Time-lapse imaging of chick spinal cord slice culture after electroporating low numbers of cells with membrane-GFP into chick.** Dorsal up. GFP, green fluorescent protein.
(MP4)

**S4 Movie. Time-lapse imaging of mouse spinal cord slice culture after electroporating low number of dorsal cells with membrane-GFP.** Dorsal to the right. GFP, green fluorescent protein.
(AVI)

**S5 Movie. Single Z-plane time-lapse imaging of chick spinal cord slice culture (from S3 Movie) after electroporating a low number of dorsal cells with membrane-GFP.** Dorsal up. GFP, green fluorescent protein.
(MP4)

**S6 Movie. Time-lapse imaging of chick spinal cord slice culture after electroporating low numbers of cells with membrane-GFP into chick.** Dorsal at one o'clock. GFP, green fluorescent protein.
(MP4)

## Acknowledgments

We thank Natalia Bulgakova for help with interpretation of EM images.

## Author Contributions

**Conceptualization:** Andrew Furley, Penny Rashbass, Marysia Placzek.

**Formal analysis:** Christine M. Tait, Kavitha Chinnaiya, Mariyam Murtaza, John-Paul Ashton, Chris J. Hill, Kate G. Storey, Marysia Placzek.

**Funding acquisition:** Jan Wijnholds, Penny Rashbass, Kate G. Storey, Marysia Placzek.

**Investigation:** Christine M. Tait, Kavitha Chinnaiya, Elizabeth Manning, Mariyam Murtaza, John-Paul Ashton, Nicholas Furley, Penny Rashbass, Kate G. Storey.

**Methodology:** Christine M. Tait, Kavitha Chinnaiya, John-Paul Ashton, Chris J. Hill, C. Henrique Alves, Jan Wijnholds, Kai S. Erdmann, Penny Rashbass, Raman M. Das, Kate G. Storey.

**Resources:** Chris J. Hill, C. Henrique Alves, Jan Wijnholds, Kai S. Erdmann, Andrew Furley, Penny Rashbass.

**Supervision:** Raman M. Das, Marysia Placzek.

**Validation:** Kavitha Chinnaiya, Raman M. Das, Marysia Placzek.

**Visualization:** Christine M. Tait, Kavitha Chinnaiya, Elizabeth Manning, Nicholas Furley, Kate G. Storey, Marysia Placzek.

**Writing – original draft:** Marysia Placzek.

**Writing – review & editing:** Christine M. Tait, Kavitha Chinnaiya, Elizabeth Manning, C. Henrique Alves, Jan Wijnholds, Kai S. Erdmann, Andrew Furley, Penny Rashbass, Raman M. Das, Kate G. Storey, Marysia Placzek.

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
