## [Editor Report · Decision Letter 0]

13 Aug 2019

Dear Dr Placzek, 

Thank you for submitting your manuscript entitled "Crumbs 2 mediates ventricular layer remodelling to form the adult spinal cord central canal" for consideration as a Research Article by PLOS Biology.

Your manuscript has now been evaluated by the PLOS Biology editorial staff as well as by an academic editor with relevant expertise and I am writing to let you know that we would like to send your submission out for external peer review.

*Please be aware that, due to the voluntary nature of our reviewers and academic editors, manuscripts may be subject to delays during the holiday season. Thank you for your patience.*

Please re-submit your manuscript within two working days, i.e. by Aug 15 2019 11:59PM.

Kind regards,

Di Jiang, PhD

Associate Editor

PLOS Biology

---

## [Decision Letter · Decision Letter 1]

11 Sep 2019

Dear Dr Placzek,

Thank you very much for submitting your manuscript "Crumbs 2 mediates ventricular layer remodelling to form the adult spinal cord central canal" for consideration as a Research Article at PLOS Biology. Your manuscript has been evaluated by the PLOS Biology editors, an Academic Editor with relevant expertise, and by twol independent reviewers.

In light of the reviews (below), we would welcome resubmission of a revised version that takes into account the reviewers' comments. You should address point 1 of reviewer 2 by providing the extended view of those movies and points 2 and 3 of this reviewer experimentally since they are key to the message of your study. We cannot make any decision about publication until we have seen the revised manuscript and your response to the reviewers' comments. Your revised manuscript is also likely to be sent for further evaluation by the reviewers.

Your revisions should address the specific points made by each reviewer. Please submit a file detailing your responses to the editorial requests and a point-by-point response to all of the reviewers' comments that indicates the changes you have made to the manuscript. In addition to a clean copy of the manuscript, please upload a 'track-changes' version of your manuscript that specifies the edits made. This should be uploaded as a "Related" file type. You should also cite any additional relevant literature that has been published since the original submission and mention any additional citations in your response. 

Before you revise your manuscript, please review the following PLOS policy and formatting requirements checklist PDF: http://journals.plos.org/plosbiology/s/file?id=9411/plos-biology-formatting-checklist.pdf. It is helpful if you format your revision according to our requirements - should your paper subsequently be accepted, this will save time at the acceptance stage.

Please note that as a condition of publication PLOS' data policy (http://journals.plos.org/plosbiology/s/data-availability) requires that you make available all data used to draw the conclusions arrived at in your manuscript. If you have not already done so, you must include any data used in your manuscript either in appropriate repositories, within the body of the manuscript, or as supporting information (N.B. this includes any numerical values that were used to generate graphs, histograms etc.). For an example see here: http://www.plosbiology.org/article/info%3Adoi%2F10.1371%2Fjournal.pbio.1001908#s5.

For manuscripts submitted on or after 1st July 2019, we require the original, uncropped and minimally adjusted images supporting all blot and gel results reported in an article's figures or Supporting Information files. We will require these files before a manuscript can be accepted so please prepare them now, if you have not already uploaded them. Please carefully read our guidelines for how to prepare and upload this data: https://journals.plos.org/plosbiology/s/figures#loc-blot-and-gel-reporting-requirements.

Upon resubmission, the editors will assess your revision and if the editors and Academic Editor feel that the revised manuscript remains appropriate for the journal, we will send the manuscript for re-review. We aim to consult the same Academic Editor and reviewers for revised manuscripts but may consult others if needed.

We expect to receive your revised manuscript within two months. Please email us (plosbiology@plos.org) to discuss this if you have any questions or concerns, or would like to request an extension. At this stage, your manuscript remains formally under active consideration at our journal; please notify us by email if you do not wish to submit a revision and instead wish to pursue publication elsewhere, so that we may end consideration of the manuscript at PLOS Biology.

When you are ready to submit a revised version of your manuscript, please go to https://www.editorialmanager.com/pbiology/ and log in as an Author. Click the link labelled 'Submissions Needing Revision' where you will find your submission record. 

Sincerely,

Di Jiang, PhD

Associate Editor

PLOS Biology

Reviewer remarks:

Reviewer #1: This manuscript describes a detailed analysis of the process of dorsal collapse in the mouse spinal cord and presents evidence that the dorsal midline radial glia trigger this collapse by producing a secreted form of Crumbs2 that induces the adjacent ventricular layer cells to lose polarity and delaminate. This is a very interesting set of results that tell an important and unexpected story, making it eminently suitable for PLoS Biology. However, there are several ways that the manuscript could be improved as outlined below:

1) As a non-mouse, non-neural developmental biologist, I found the initial description of the cell-types involved in dorsal collapse very hard to follow. It would help a lot to have a well-labelled diagram in figure 1

2) The EM images in figure 4 are difficult to interpret as the boundaries between cells are not clear. This makes it a matter of trust for the reader to believe that the green arrows mark junctions between cells. This is complicated by the fact that the green arrows only mark the junctions that are not apparent and the prominent junctions are not labelled.

3) The authors argue that Crb2S disrupts epithelial polarity in the ventricular layer, which leads to delamination, but it is not clear from their data that this is the correct causal connection. In figure 4, ZO-1 and aPKC disappear from a number of VL cells, not just those that are delaminating. In addition, the cells appear to delaminate by apical abscission, which is a completely different mechanism to a loss of the apical domain. 

4) The effects of Crb2S on MDCK cells would be more compelling if the untreated cells were polarised, but this is hard to see in the images shown.

5) The Crb2 knockout in the Nestin +ve cells gives a very convincing phenotype, but left me confused. Is the Cre recombinase expressed only in the dmNes+RG cells, which are the Nestin positive cells visible in figure 2, or is it expressed in the whole ventricular layer as stated in the text? If the latter is the case, this does not prove a role for Crb2S in delamination as the effect could be due to the loss of the transmembrane form in the VL cells. However, this would also show that loss of apical Crb2 and aPKC from the VL cells does not lead to a loss of polarity or delamination. Please clarify.

6) The authors propose in the discussion that loss of Crb2 and aPKC promotes the formation of an apical actin cable in the VL cells, but this does not seem consistent with their data. The dorsal VL cells with reduced Crb2 and aPKC appear to have less apical F-actin than other cells, and this does not form the straight line that one would expect for a supracellular actin able.

Reviewer #2: This paper investigates the mechanism of ‘dorsal collapse’ whereby the originally extensive neural tube lumen is converted through development to a much smaller, ventrally located lumen. Concomitantly, the ependymal cell population becomes defined. This phenomenon is well described and is known to involve transformation of radial glia and a key role of apical proteins in the neuroepithelium. The authors investigate the mechanism further in mouse and chick, and obtain evidence for a non-cell-autonomous role of a secreted form of the Crumbs2 protein.

The paper is interesting and the studies have been carefully performed. The work is well described and documented in the main and supplementary figures. The paper divides into two parts. The first is a molecular anatomical study, in which a key role for the nestin-positive radial glia is mainly studied. I would say this is valuable in extending existing knowledge on dorsal collapse. The second part is experimental, involving testing of the hypothesis that the CRB2S secreted variant plays an essential role in dorsal collapse. This part of the study is innovative and has the potential to significantly advance this research field. However, I feel the experimental studies currently fall short of providing convincing evidence to support the CRB2S hypothesis.

1. Ratcheting is proposed as a mechanism of dorsal collapse. Live imaging of sparsely transfected spinal cord slice cultures provides evidence for this. In Fig 3A, I can clearly see the second dorsal VL progenitor cell making contact with the dmNes+RG cell, in frames 990-1060, after apical abscission of the previous contacting cell. Interestingly, Movie 1 seems to stop before this critical event - is it missing the last frames? In any case, how do the authors know that after abscission of a dorsal VL progenitor, there is “ratcheting down” so that the next one (ventrally) then makes contact? I cannot really see convincing evidence for this conclusion. For example, is there a movie in which a labelled (and so clearly more ventral) VL cell becomes connected after a clearly more dorsal one has abscised?

2. The main effect of dmNes+RG cell transplantation into the chick neural tube lumen seems to be breakage of the basement membrane, with cells becoming located outside the normal confines of the spinal cord. A similar effect is observed after implantation of beads soaked in purified CRB2S. However, this basement membrane breakage is not normally observed during dorsal collapse. Perhaps the authors would say their manipulations deliver abnormally large amounts of CRB2S, that have additional effects in the spinal cord? However, I am struggling to see much resemblance between the normal dorsal collapse process and the changes induced by delivering CRB2S. Can the effect be titrated, by implanting beads soaked in smaller amounts of CRB2S? Alternatively, perhaps this gain-of-function experiment is problematic, as the recipient spinal cord is already ‘dorsally collapsing’ and so the additional effects of CRB2S release are not easy to interpret. Could the authors introduce a blocking antibody to CRB2S that would serve as a loss-of-function experiment, to see if dorsal collapse is diminished?

3. The mouse knockout of Crb2 does indeed produce faulty dorsal collapse, consistent with the authors’ hypothesis. However, one could argue that this is happening because of the need for Crb2 for the vital regulation of apico-basal polarity in the neuroepithelium, and not specifically through interference with the postulated CRB2S mechanism. An obvious further experiment is to genetically prevent CRB2S production, perhaps by introducing a specific mutation to prevent the causative splicing event, using CRISPR technology. Without such a more specific experiment, I think the mouse knockout does not provide strong evidence for the authors’ hypothesis.

Minor points:

4. Fig 1. Diagrams at the bottom are not referred to in the legend.

5. Fig 2. Sentence: “(EH) Immunolabelling reveals ventral and dmNes+RG that elongate as collapse proceeds …” does not make sense. Please explain what red line indicates in part I, and mention/give labels to the diagrams to make clear which data each is summarising.

6. Fig 3. Title mentions ‘ratcheting’, but it is not clear from reading the legend what is the evidence in the data for this phenomenon. The link to ratcheting needs to be made (see also point 1 above).

7. Fig 5. Please explain in the legend what M2 means. The site of the beads in parts O,P,Q,R should be shown (e.g. by dotted lines). Similarly in Suppl Fig 8. Is the asterisk in D indicating a bead? Please explain in the legend.

8. “High power” is used throughout to refer to images. Presumably the authors mean “high magnification”?

9. The paper title within the reference Sevc et al, 2009 is incorrect.

---

## [Decision Letter · Decision Letter 2]

22 Jan 2020

Dear Dr Placzek,

Thank you for submitting your revised Research Article entitled "Crumbs 2 mediates ventricular layer remodelling to form the spinal cord central canal" for publication in PLOS Biology. I have now obtained advice from the original reviewers and have discussed their comments with the Academic Editor. 

Based on the reviews, we will probably accept this manuscript for publication, assuming that you will modify the manuscript to address the remaining points raised by the reviewers. We would like to ask you to add a discussion about the study by Kuriyama et al. on Xerl. Please also make sure to address the data and other policy-related requests noted at the end of this email.

We expect to receive your revised manuscript within two weeks. Your revisions should address the specific points made by each reviewer. In addition to the remaining revisions and before we will be able to formally accept your manuscript and consider it "in press", we also need to ensure that your article conforms to our guidelines. A member of our team will be in touch shortly with a set of requests. As we can't proceed until these requirements are met, your swift response will help prevent delays to publication.

Sincerely,

Di Jiang

PLOS Biology

ETHICS STATEMENT:

Please include an Ethics Statements as a separate subsection in the beginning of the Methods section of your manuscript.

-- Please include the full name of the IACUC/ethics committee that reviewed and approved the animal care and use protocol/permit/project license. Please also include an approval number.

-- Please include the specific national or international regulations/guidelines to which your animal care and use protocol adhered. Please note that institutional or accreditation organization guidelines (such as AAALAC) do not meet this requirement.

-- Please include information about the form of consent (written/oral) given for research involving human participants. All research involving human participants must have been approved by the authors' Institutional Review Board (IRB) or an equivalent committee, and all clinical investigation must have been conducted according to the principles expressed in the Declaration of Helsinki.

DATA POLICY:

Regardless of the method selected, please ensure that you provide the individual numerical values that underlie the summary data displayed in the following figure panels as they are essential for readers to assess your analysis and to reproduce it: Figure 1EJO, 6FG. NOTE: the numerical data provided should include all replicates AND the way in which the plotted mean and errors were derived (it should not present only the mean/average values).

Reviewer remarks:

Reviewer #1: This revised manuscript has addressed most of the issues raised by the referees and I think that the results are now suitable for publication. I still have two concerns tht it would be good if they could address:

1) The discussion of the mechanism of CRB2S action is rather confused. On the one hand, the authors state that 

"however, dmNes+RG also secrete CRB2S, which acts non cell-autonomously and locally to compete away CRB2 in neighboring cells, leading to loss of apico-basal polarity. Together these findings add to the evidence that the fly regulatory network is conserved in vertebrates."

On the other hand, they show that loss of all CRB2 isoforms in the VL cells does not lead to a loss of polarity or delamination and conclude that

"Our studies suggest that the CRB2S-CRB2-mediated polarity changes exert

specific downstream effects to enable delamination, which are not triggered

simply through genetic loss of full-length CRB2"

This seems to indicate to me that CRB2 is not required for polarity in the VL cells and that CRB2S must signal through some other mechanism. They are also confused in their description of the literature of CRB2 stating that: 

disruption of CRB2-CRB2 leads to a loss of polarity and disrupted epithelial-mesenchymal transition (EMT)

 Ramkumar showed that Crumbs is essential in the primitive streak for EMT (which is a loss of epithelial polarity), so this is the opposite of its disruption leading to a loss of polarity. I suspect that that they are trying to shoehorn their results to fit the consensus model that CRB2 is required for polarity in vertebrate epithelia as it is in flies, whereas their data and Ramkunar's data suggest the opposite, i.e that it is required for a loss of polarity and EMT.

2) I still find the MDCK data unsatisfactory. The effect on adhesion is clear, but the loss of polarity is not really obvious.

Reviewer #2: The authors have greatly improved the manuscript in the revised version, and are to be congratulated on providing the additional evidence for the action of CRB2S from the new blocking antibody and shCrb2 implantation studies. Additionally, Fig 3 showing the movie frames of delamination of VL cells relative to the dmNes+RG cell is now very much clearer than the corresponding figure in the original version.

I would only now request some final improvements to the documentation of the findings and experimental procedures, in the figure legends and Methods, as follows:

Fig 2. The schematics on the left and in the centre are not give labels or specifically explained in the legend. Please make this change.

Fig 2E-H. This magnification seems too low to clearly show individual Nestin+ cells. Please indicate by arrows where the cells are on the sections, which the legend refers to.

Fig 2J-L''. Please explain in the legend what the dotted lines and arrowheads refer to.

Fig 2N. Please explain what the insets of boxed regions show. I do not think the "dorso-ventrally oriented Sox2(+) nuclei" are clear without arrows.

Fig 5H-L. The Results comments on "… Nkx6.1(+) progenitor cells appeared disorganised, the basement membrane showed breaks …". Please indicate these by arrows in the figure and include mention of these in the legend.

Fig 7. Part U is not labelled as such.

Methods. I think more detail is needed of the specialised in vivo procedure that has been done. The authors say they grafted dmNes+RG or VL cells, but there is very little detail on how such cells were isolated in pure form for grafting. The Methods says: "Tissue to be transplanted was punched out with a pulled glass needle (1mm x 0.78, Harvard Apparatus) and mouth pipette (Sigma), before being carefully placed into the most rostral part of the open neural tube." This leaves unclear what evidence the authors have that their punching out method was able to harvest dmNes+RG or VL cells, free from contamination with each other. Were tissue punches evaluated after isolation with markers to test this? Better description and/or more evidence of this is needed.

---

## [Editor Report · Decision Letter 3]

18 Feb 2020

Dear Dr Placzek,

On behalf of my colleagues and the Academic Editor, Catherina Becker, I am pleased to inform you that we will be delighted to publish your Research Article in PLOS Biology. 

Early Version

PRESS 

Kind regards,

Alice Musson

Publication Assistant, 

PLOS Biology

on behalf of

Di Jiang,

Associate Editor

PLOS Biology